EMBO
Molecular Medicine

# Bcl-xL targeting eliminates ageing tumor-promoting neutrophils and inhibits lung tumor growth

Anita Bodac[1], Abdullah Mayet[2,3,4], Sarika Rana[2,3,4], Justine Pascual[5], Amber D Bowler [ID][1], Vincent Roh[6,7], Nadine Fournier [ID][6,7], Ligia Craciun[8], Pieter Demetter[8], Freddy Radtke [ID][1] & Etienne Meylan [ID][2,3,4 ✉]

## Abstract

**Elevated peripheral blood and tumor-infiltrating neutrophils are often associated with a poor patient prognosis. However, therapeutic strategies to target these cells are difficult to implement due to the life-threatening risk of neutropenia. In a genetically engineered mouse model of lung adenocarcinoma, tumor-associated neutrophils (TAN) demonstrate tumor-supportive capacities and have a prolonged lifespan compared to circulating neutrophils. Here, we show that tumor cell-derived GM-CSF triggers the expression of the anti-apoptotic Bcl-xL protein and enhances neutrophil survival through JAK/STAT signaling. Targeting Bcl-xL activity with a specific BH3 mimetic, A-1331852, blocked the induced neutrophil survival without impacting their normal lifespan. Specifically, oral administration with A-1331852 decreased TAN survival and abundance, and reduced tumor growth without causing neutropenia. We also show that G-CSF, a drug used to combat neutropenia in patients receiving chemotherapy, increased the proportion of young TANs and augmented the anti-tumor effect resulting from Bcl-xL blockade. Finally, our human tumor data indicate the same role for Bcl-xL on pro-tumoral neutrophil survival. These results altogether provide preclinical evidence for safe neutrophil targeting based on their aberrant intra-tumor longevity.**

**Keywords** Tumor-associated Neutrophils; Bcl-xL; Lung Adenocarcinoma; Mouse Models of Lung Cancer
**Subject Categories** Cancer; Immunology; Respiratory System

## Introduction

Neutrophils are the most abundant leukocytes present in the blood and function as a first-line defense against pathogens (Kobayashi and DeLeo, 2009; Hidalgo and Casanova-Acebes, 2021). Despite their usually short lifespan, it is now recognized that neutrophils participate in the establishment of chronic diseases such as cancer and support the spread of metastases to distant tissues (Gentles et al, 2015; Shaul and Fridlender, 2019; Coffelt et al, 2015; Yang et al, 2020). In non-small cell lung cancer (NSCLC), a high neutrophil-to-lymphocyte (NLR) ratio and high density of intra-tumoral neutrophils are prognostic for poor patient survival (Kargl et al, 2017; Faget et al, 2017). The tumor-promoting functions of neutrophils are many and versatile (Jaillon et al, 2020; Siwicki and Pittet, 2021). They can, for instance, stimulate tumor progression through the production of reactive oxygen species (ROS) (Canli et al, 2017; Zhong et al, 2021) or the secretion of angiogenic factors (Scapini et al, 2004; Nozawa et al, 2006; Kuang et al, 2011). In parallel, they can prevent the action of cytotoxic T cells through the secretion of arginase, ROS and nitric oxide (NO) (Munder et al 2006; Mensurado et al, 2018; Casbon et al, 2015), thus creating a favorable environment for tumor growth.

Strategies to deplete neutrophils in vivo, while providing important biological information in laboratory models (Boivin et al, 2020), are not translatable to humans as they could induce neutropenia, a major life-threatening condition where the body is unable to fight infections. For this reason, neutrophils have not been considered promising targets for treating cancer. There is now accumulating evidence, however, for molecular and phenotypic diversity of neutrophils in solid tumors. In a mouse model of lung cancer, analysis of the neutrophil compartment revealed that a subset of cells, marked by surface expression of the sialic acid-binding protein SiglecF, exhibit pro-tumor activity (Engblom et al, 2017; Pfirschke et al, 2020), suggesting that tumors can re-educate these cells for their benefit. We now know that there is great neutrophil heterogeneity in NSCLC (Zilionis et al, 2019a; Salcher et al, 2022), which should allow the identification of targeted therapies that disrupt selected tumor-associated neutrophils (TANs) without impacting the basal pool of circulating neutrophils, necessary for the host innate immune defense.

Physiologically, neutrophils patrol the organism for a short time before their death and clearance; homeostasis is maintained by daily replenishment of neutrophils from the bone marrow. This ensures an efficient set of pathogen-killing cells, while at the same

---

[1]Swiss Institute for Experimental Cancer Research, School of Life Sciences, Ecole Polytechnique Fédérale de Lausanne, 1015 Lausanne, Switzerland. [2]Laboratory of Immunobiology, Department of Molecular Biology, Université libre de Bruxelles, 6041 Gosselies, Belgium. [3]Lung Cancer and Immuno-Oncology laboratory, Bordet Cancer Research Laboratories, Institut Jules Bordet, Hôpital Universitaire de Bruxelles, Université libre de Bruxelles, 1070 Bruxelles, Belgium. [4]ULB Cancer Research Center (U-CRC) and ULB Center for Research in Immunology (U-CRI), 1070, Bruxelles, Belgium. [5]Center for Integrative Genomics, Faculty of Biology and Medicine, University of Lausanne, 1015 Lausanne, Switzerland. [6]Translational Data Science - Facility, SIB Swiss Institute of Bioinformatics, 1015 Lausanne, Switzerland. [7]Agora Cancer Research Center, 1005 Lausanne, Switzerland. [8]Department of Pathology, Institut Jules Bordet, Hôpital Universitaire de Bruxelles, Université libre de Bruxelles, 1070 Bruxelles, Belgium. ✉E-mail: etienne.meylan@ulb.be

time preventing the deleterious effects they could have on healthy tissue, such as in the establishment of chronic inflammation (Hidalgo and Casanova-Acebes, 2021b). Nevertheless, it has been shown that neutrophil death is delayed within inflamed tissues upon secretion of inflammatory mediators (Fox et al, 2010). Surprisingly, we and others have recently shown that SiglecF[+] neutrophils survive within the lung tumor microenvironment for more than 6–8 days (Ancey et al, 2021a; Pfirschke et al, 2020; Boivin et al, 2021), suggesting that these cells undergo profound intrinsic changes when exposed to the tumor microenvironment, enabling adaptation and prolonged survival. For example, a metabolic shift toward increased glucose usage, triggered by glucose transporter Glut1 upregulation in TANs, contributes to their ageing and tumor support (Ancey et al, 2021a; Bodac and Meylan, 2021).

Apoptosis is an extensively studied, regulated form of cell death that is triggered and modulated through two main axes known as the extrinsic and intrinsic pathways (Galluzzi et al, 2018). The latter is tightly regulated by the balance of pro- and anti-apoptotic proteins of the B cell lymphoma protein 2 (Bcl-2) family (Hatok and Racay, 2016), amongst which Bcl-xL has an anti-apoptotic role. It binds to the pro-apoptotic effectors Bak and Bax, preventing their clustering and the permeabilization of the mitochondrial outer membrane, which otherwise leads to the release of cytochrome c and downstream activation of caspases. In cancer, Bcl-xL was linked to tumor cell proliferation and resistance to chemotherapy, including in lung cancer (Stover et al, 2019; Shen et al, 2018). While the expression of Bcl-xL in neutrophils remained controversial for some time, recent studies show an induction of the protein in inflammatory conditions where neutrophil apoptosis is delayed, like sepsis (Guo et al, 2006) and rheumatoid arthritis (Carrington et al, 2021).

Here, we demonstrate that TANs' death is prevented by GM-CSF-dependent Bcl-xL induction. A-1331852, an orally available, potent, and highly selective Bcl-xL inhibitor, reduces TAN survival and burden while sparing peripheral blood neutrophils in the $Kras^{Frt-STOP-Frt-G12D/WT}$; $p53^{Frt/Frt}$ (hereafter called KP) mouse model of lung adenocarcinoma (LUAD), and its long-term use diminishes lung tumor growth.

# Results

## Anti-apoptotic Bcl-xL is induced in TANs

As the longevity of neutrophils extends within the tumor mass in the KP mouse model of LUAD, we analyzed our previously generated bulk RNA sequencing data (data ref: Ancey et al, 2021b) to query genes that could modulate survival or apoptosis. In the comparison between healthy lung neutrophils (HLNs) and TANs, we found that the apoptosis gene set was downregulated in the latter, suggesting that the tumor microenvironment stimulates the escape of TANs from apoptosis (Fig. EV1A). Monitoring the expression of anti-apoptotic genes, $Bcl2l1$ (coding for Bcl-xL) was upregulated in TANs compared to HLNs. This was also the case for $Bcl2a1b$ (BFL-1), albeit with a lower significance, while $Mcl1$ was more expressed in HLNs and $Bcl-2$ was not detected, probably due to low expression in these cells (Fig. EV1B). Moreover, by real-time PCR analysis, $Bcl2l1$ expression—but not that of the other Bcl-2

family members—was significantly higher in TANs versus HLNs (Fig. EV1C). Collectively, these analyses, together with the availability of small molecule inhibitory compounds, prompted us to investigate Bcl-xL further. Immunofluorescence staining showed colocalization of myeloperoxidase (MPO), a marker of neutrophils, and Bcl-xL in KP tumors, validating the expression of the protein in TANs in vivo (Fig. EV1D).

To determine whether Bcl-xL expression is triggered in neutrophils upon arrival in the tumor mass or before, we compared Bcl-xL levels from healthy or tumor-bearing mice, in neutrophils extracted from the bone marrow (BMN), peripheral blood (PBN), lung (HLN) or tumors (TAN), using flow cytometry. All neutrophil populations expressed Bcl-xL, but the levels were elevated in HLNs and the highest in TANs, suggesting tissue-specific induction of Bcl-xL (Fig. 1A–C).

Six different neutrophil subsets (N1-N6) have been identified by single-cell RNA sequencing in an orthotopic mouse model of lung cancer (Zilionis et al, 2019a). Amongst them, N4, which was characterized by high $SiglecF$, also displays high $Bcl2l1$ (and $Bcl2a1b$) gene expression compared to the other subsets. In contrast, $Bcl-2$ was almost not expressed, and $Mcl1$ was highly expressed in most subsets, but particularly in N1 (Fig. EV1E). Following this observation, we wanted to explore if SiglecF[+] TANs, a tumor-supportive subset of cells that are longer lived than SiglecF[−] TANs (Pfirschke et al, 2020), were expressing higher levels of the anti-apoptotic protein in autochthonous KP tumors. Using flow cytometry, we measured higher Bcl-xL levels in SiglecF[+] TANs compared to their SiglecF[-] counterparts (Fig. 1D), suggesting that Bcl-xL supports TAN differentiation toward pro-tumorigenic SiglecF[+] cells.

We then hypothesized that the tumor cell secretome triggers the extended survival of TANs. To test this, we incubated BMNs with conditioned medium from KP tumors or from SV2, a cell line derived from a $Kras^{Lox-STOP-Lox-G12D/WT}$; $p53^{Flox/Flox}$ tumor (see scheme Fig. 1E). We measured by flow cytometry the percentage of neutrophils alive after 24 h of incubation (Fig. EV1F) and determined that tumor (or SV2)-derived supernatant enhanced neutrophil survival by twofold compared to neutrophils cultured with medium only (Fig. 1F). Real-time PCR analysis revealed an increased $Bcl2l1$ gene expression in BMNs incubated with tumor cell supernatant, which was confirmed at the protein level by western blot (Fig. 1G,H).

These data demonstrate the induction of the anti-apoptotic Bcl-xL protein in TANs, which can be recapitulated in vitro when neutrophils are cultured with a tumor cell-derived supernatant.

## Tumor cells induce Bcl-xL in neutrophils via GM-CSF-mediated JAK/STAT signaling

During inflammation, the death of neutrophils can be delayed in response to several signals, such as growth factors, cytokines, or danger-associated motifs (Simon, 2003; Gabelloni et al, 2013), with different signaling pathways reported. In the cancer context, to determine which pathway is involved, we decided to incubate freshly isolated BMNs with SV2 supernatant and increasing doses of selected pathway inhibitors, including ruxolitinib (JAK1/2 inhibitor), stattic (STAT3 inhibitor), MLN120B (IKKβ inhibitor) and Ly294002 (PI3K inhibitor). After 24 h of incubation, each of stattic and ruxolitinib inhibited neutrophil survival, while the other

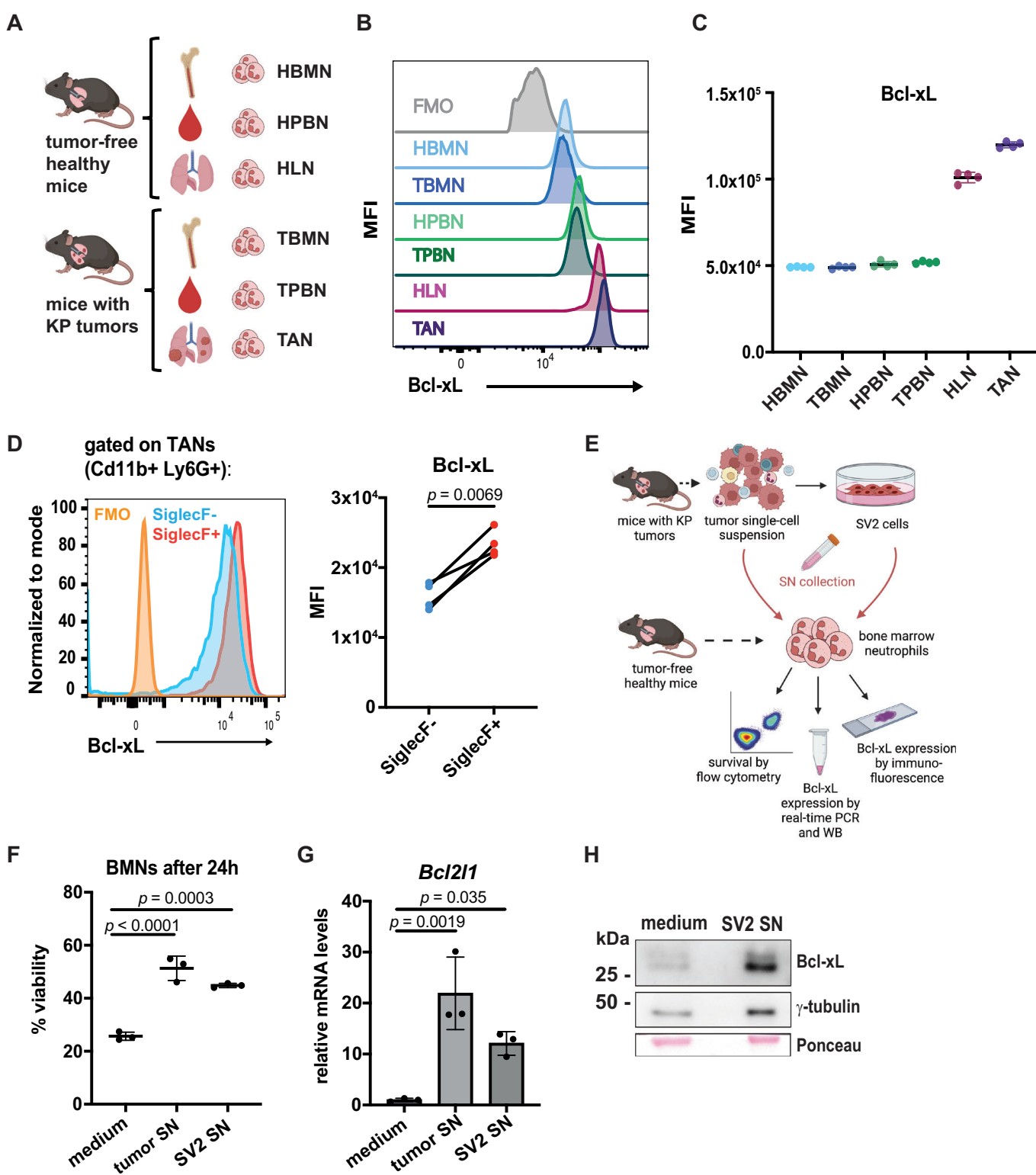

two compounds had no effect (Figs. 2A and EV2A). In the same conditions, both stattic and ruxolitinib repressed Bcl-xL in a dose-dependent manner (Fig. 2B). These results altogether show that the tumor supernatant expands neutrophil survival and induces Bcl-xL expression through the JAK-STAT pathway. Of note, stattic and ruxolitinib did not diminish basal neutrophil survival, confirming that JAK-STAT signaling is selectively triggered when neutrophils are incubated with a tumor cell-derived supernatant (Fig. EV2B).

Remotely, cancer cells can skew hematopoiesis toward increased myeloid cell production by releasing factors such as G-CSF and

**Figure 1. Tumor-associated neutrophils overexpress the anti-apoptotic protein Bcl-xL.**

(A) Scheme describing the different neutrophil populations analyzed. For healthy naive mice: bone marrow-derived neutrophils (HBMN), healthy peripheral blood neutrophils (HPBN), healthy lung neutrophils (HLN). For tumor-bearing mice: bone marrow-derived neutrophils (TBMN), peripheral blood-derived neutrophils (TPBN) and tumor-associated neutrophils (TAN). (B) Representative flow cytometry histograms showing Bcl-xL levels in different neutrophil populations in naive mice and mice with KP tumors. (C) Median fluorescence intensity (MFI) of Bcl-xL in indicated populations ($n = 4$ biological replicates). (D) Representative flow cytometry histograms showing Bcl-xL levels in SiglecF$^-$ and SiglecF$^+$ TANs and corresponding MFI quantification ($n = 4$). (E) Scheme representing the experimental setup for the neutrophil survival assay. (F) Percentage of viable neutrophils after 24 h in medium, tumor-derived supernatant (SN) or SV2 cell-derived SN ($n = 3$ biological replicates). (G) Real-time PCR analysis of *Bcl2l1* in BMNs upon incubation with medium or supernatant from tumors or from SV2 cells. Data are shown as mean ± SD of $n = 3$ mice of a representative experiment reproduced three times. (H) Western blot analysis of Bcl-xL expression in BMNs after 24 h incubation in medium or SV2 SN. γ-tubulin and Ponceau red were used as loading control. Data information: In vitro assays were performed at least three times. All results are shown as mean ± SD. and statistical analysis was performed using paired *t* test (D) or one-way ANOVA (F). For (F), the experiment was replicated three times. For (G), significance was determined by ordinary one-way ANOVA with Dunnett's multiple comparisons test. Source data are available online for this figure.

GM-CSF (Casbon et al, 2015; Bayne et al, 2012). With an analysis of the single-cell transcriptomics data (data ref: Zilionis et al, 2019b), there was a predominant G-CSF receptor (*Csf3r*) expression in neutrophils from healthy mice. In contrast, the N4 SiglecF$^+$ TAN subset showed a lower *Csf3r* expression with an increased expression of *Csf2rb* and *Csf2rb2*, which encode one of the chains of the heterodimeric GM-CSF receptor (GM-CSFR); the gene encoding the other chain, *Csf2ra*, was expressed in all neutrophil subsets (Fig. EV2C). This observation was interesting, as GM-CSF is a known activator of the JAK-STAT pathway (Al-Shami et al, 1998). Thus, we hypothesized that GM-CSF sustains neutrophils in the lung tumor microenvironment and supports their transition into SiglecF$^+$ cells. First, we tested if GM-CSF was able to induce Bcl-xL expression. We cultured BMNs with 10 ng/mL of GM-CSF for 24 h, which increased their survival to a similar extent as the tumor cell supernatant did (Fig. 2C). Importantly, both were prevented upon the addition of 5 μM stattic, indicating GM-CSF-induced neutrophil survival is mediated by the JAK-STAT signaling pathway (Fig. 2C). Western blot analysis confirmed the upregulation of Bcl-xL in BMNs incubated with GM-CSF, which was abrogated by stattic (Fig. 2D). Finally, neutralization with anti-GM-CSF counteracted the tumor supernatant-dependent increased neutrophil viability (Fig. 2C). Together, these data position GM-CSF as a major contributor for neutrophil survival mediated by a supernatant of in vitro cultured lung tumor cells.

Simultaneously, we wanted to measure if GM-CSF derived from the supernatant could induce SiglecF expression in cultured bone marrow neutrophils. After 24 h of incubation, both the tumor cell supernatant and GM-CSF induced cell surface SiglecF expression in BMNs, which was diminished with increasing doses of stattic (Fig. 2E). Interestingly, a tumor cell supernatant previously incubated with the neutralizing GM-CSF antibody showed a reduced proportion of SiglecF-expressing cells, demonstrating that GM-CSF is necessary and sufficient for SiglecF induction in isolated BMNs (Fig. EV2D).

We then investigated the relationship between GM-CSF and SiglecF by quantifying GM-CSF in single KP tumors with enzyme-linked immunosorbent assay (ELISA). We determined a positive correlation between GM-CSF and the absolute numbers of total TANs ($R^2 = 0.42$), which was weaker and stronger when considering SiglecF$^-$ ($R^2 = 0.39$) and SiglecF$^+$ TANs ($R^2 = 0.47$), respectively (Fig. 2F). We also measured GM-CSF from the blood and the bronchoalveolar lavage fluid (BALF) of healthy and tumor-bearing mice. We were not able to detect GM-CSF in the blood from any subject and hardly detected it in the BALF of healthy individuals. However, three out of four tumor-bearing mice showed elevated

levels of this growth factor in the BALF (Fig. EV2E). In human LUAD, *CSF2* (gene coding for GM-CSF) expression correlated with poorer overall survival (Fig. EV2F). Altogether, these data suggest that local GM-CSF secretion, possibly by tumor cells, mediates TAN polarization toward SiglecF$^+$, long-lived and tumor-supportive cells in vivo.

## Bcl-xL blockade impairs TAN ageing

Having established the link between Bcl-xL induction and neutrophil ageing, we decided to investigate if Bcl-xL blockade could prevent neutrophil survival in vitro and reduce TANs' lifespan in vivo. For this, we selected two known BH3 mimetic drugs tested in clinical studies, Navitoclax, a Bcl-2/Bcl-xL dual inhibitor (Tse et al, 2008; Mohamad Anuar et al, 2020) and Venetoclax (Souers et al, 2013; Lasica and Anderson, 2021), a Bcl-2-specific inhibitor (Fig. EV3A). We also chose A-1331852, a more recently developed BH3 mimetic that binds to Bcl-xL with strong affinity and inhibits it with high selectivity (Wang et al, 2020; Leverson et al, 2015). Of note, A-1331852 has a stronger affinity for Bcl-xL than Navitoclax and is comparatively more potent to impair the growth of Bcl-xL-dependent tumor cells (Leverson et al, 2015).

After incubating BMNs with SV2 cell-derived supernatant together with increasing doses of each inhibitor, we noticed that A-1331852 diminished neutrophil survival even at the lowest dose of 0.1 nM. In contrast, Navitoclax and Venetoclax only partially reduced neutrophil survival at the highest dose (100 nM) (Fig. EV3B), showing that Bcl-xL but not Bcl-2 mediates neutrophil survival in tumor supernatant conditions. Importantly, A-1331852 did not affect neutrophils cultured in normal medium (Fig. 3A), suggesting a window of opportunity for targeting aberrantly ageing TANs while sparing normal neutrophils. Using TNF as positive control, we measured that neutrophil apoptosis was inhibited by the tumor cell supernatant while it was enhanced by A-1331852, as inferred from both Annexin V/7-AAD staining by flow cytometry and cleavage of caspase-3 by western blot (Fig. EV3C). Like BMNs, the survival of HLNs increased with the supernatant, which was inhibited by the addition of A-1331852 (Fig. EV3D). To test if TANs were sensitive to the drug, we purified them from KP tumors and cultured them with the same doses as for BMNs, in normal- or in tumor cell supernatant. As opposed to BMNs, A-1331852 induced TAN cell death already in basal conditions, demonstrating that Bcl-xL induction in TANs renders them vulnerable to its blockade (Fig. 3B). In SV2 cell supernatant, TANs remained sensitive to A-1331852 (Fig. 3B), while their pre-incubation with

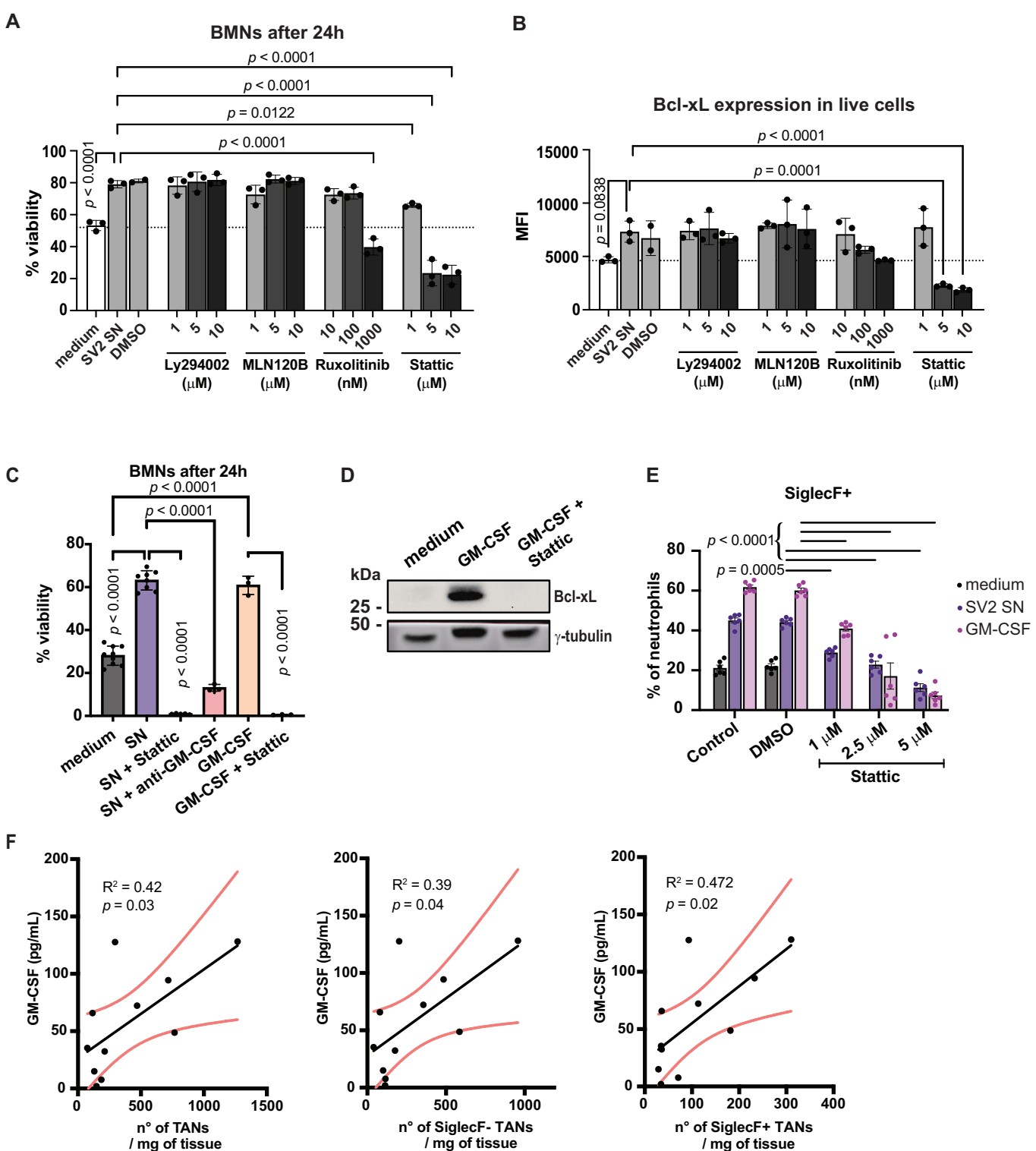

Z-VAD-FMK, a pan-caspase inhibitor, counteracted the effect of A-1331852 (Fig. EV3E). Finally, to directly compare the cytotoxic activity of Bcl-xL blockade on TAN subsets, we incubated freshly isolated TANs with an anti-SiglecF antibody immediately prior to a 6-h incubation with A-1331852. This experiment demonstrated a specific killing of SiglecF$^+$ TANs (Fig. 3C), highlighting the

vulnerability of this tumor-supportive subset in comparison to other TANs.

We next wanted to monitor if A-1331852 could interfere with TAN ageing in vivo when administered to KP tumor-bearing mice. To separate TANs based on their age, we injected mice with BrdU, a thymidine analog that incorporates into the DNA of proliferating

**Figure 2.  Bcl-xL is induced by GM-CSF-mediated JAK-STAT signaling.**

(A) Percentage of viable bone marrow neutrophils (BMN) after 24 h of incubation with medium, SV2 cell supernatant (SN), DMSO as control or SN with the indicated doses of Ly294002 (PI3K inhibitor), MLN120B (IKKβ inhibitor), ruxolitinib (JAK1/2 inhibitor) or stattic (STAT3 inhibitor). The mean basal neutrophil survival (incubated with medium) is indicated with a dashed line. (B) Mean fluorescence intensity (MFI) of Bcl-xL expression in neutrophils from (A). The mean basal intensity is indicated with a dashed line. (C) Percentage of viable BMNs after 24 h of incubation. (D) Western blot analysis of Bcl-xL from BMNs incubated with GM-CSF (10 ng/mL) and stattic (5 μM) for 24 h. γ-tubulin was used as a loading control. (E) SiglecF surface protein expression measured in BMNs ($n = 6$ biological replicates) incubated with SV2 SN or GM-CSF and with or without stattic at the indicated concentrations for 24 h. (F) Pearson correlation analysis between the frequency of TANs (gated on CD45$^+$CD11b$^+$Ly6G$^+$ cells), SiglecF$^-$ TANs and SiglecF$^+$ TANs and GM-CSF levels in individual tumors ($n = 11$). Data information: Data are shown as mean ± SD. For (A, B), data represent BMNs from $n = 3$ mice. For (C), data for control ($n = 9$), SN ($n = 8$), and SN + stattic ($n = 5$) are pooled from three independent experimental groups, and for the others, $n = 3$ represents three biological replicates. Significance was determined using ordinary one-way ANOVA with Dunett's (A, B) or Tukey's (C) multiple comparison test. For (E), significance was determined using two-way ANOVA with Tukey's multiple comparisons test. Source data are available online for this figure.

cells, including neutrophil progenitors in the bone marrow. As mature neutrophils are post-mitotic, BrdU will remain detectable in these cells until they die (Ancey et al, 2021a). Knowing that TANs can survive for more than 6 days within the lung tumor microenvironment (Ancey et al, 2021a), we chose to inject KP mice with BrdU and to start the treatment the following day with either Venetoclax or Navitoclax for 6 days. Endpoint analysis revealed no difference in total TAN and SiglecF$^+$ TAN prevalence upon each treatment. However, we observed a significant reduction in SiglecF$^+$BrdU$^+$ TANs, i.e., in 6.5-day-old TANs from mice treated with Navitoclax, compared to the control and Venetoclax-treated groups (Fig. EV3F). These results suggest a contribution for Bcl-xL but not Bcl-2 in sustaining neutrophil survival within tumors.

To confirm the importance of Bcl-xL in TAN survival, we injected tumor-bearing mice with a single dose of BrdU and started A-1331852 treatment the next morning for a duration of 8 days before sacrifice (see scheme Fig. 3D). Flow cytometry analysis revealed that there was no difference in total TAN prevalence (Fig. 3D). However, qualitative differences in TAN subsets occurred: SiglecF$^+$ TANs were reduced in treated mice compared to controls, which was attributed to a very significant decrease of the 8.5-day-old SiglecF$^+$BrdU$^+$ cells (Fig. 3D). The loss of this old TAN subset was accompanied by a gain of younger, SiglecF$^-$BrdU$^-$ TANs, highlighting an old-to-young TAN shift upon short-term A-1331852 treatment. Correspondingly, *Siglecf* expression, measured by real-time PCR analysis from total TANs, was reduced upon Bcl-xL blockade (Fig. EV3G). Recently, a TAN subset distinct from SiglecF$^{high}$ cells and marked by an interferon-stimulated gene (ISG) signature was identified as essential for the successful response to anti-CD40 in orthotopic *Kras*$^{Lox-STOP-Lox-G12D/WT}$; *p53*$^{Flox/Flox}$ tumors (Gungabeesoon et al, 2023). In contrast to *Siglecf*, A-1331852 did not diminish the expression of interferon-related genes and ISGs by TANs; it even increased *Cxcl10* and *Ddx58* (Fig. EV3G), suggesting that Bcl-xL blockade selectively eliminates SiglecF$^{high}$ neutrophils and preserves ISG-expressing TANs.

To interrogate the consequences of Bcl-xL inhibition on the lung tumor immune microenvironment globally, we treated mice with A-1331852 for 2 weeks and performed a full-spectrum flow cytometry analysis of tumors (Appendix Fig. S1). Our data revealed trends toward fewer neutrophils and more CD3$^+$ T cells upon treatment (Fig. 3E). Looking more in-depth at the neutrophil population, we found that SiglecF$^+$ TANs were significantly decreased, which was accompanied by a decrease in PD-L1-expressing TANs, as most of them were SiglecF$^+$ (Fig. 3E). Of note, Bcl-xL expression—higher in SiglecF$^+$ than in SiglecF$^-$ TANs— significantly diminished in each subset after treatment (Fig. EV3H),

suggesting a preferential killing of Bcl-xL high expressors. As for T lymphocytes, we observed an almost significant increase in CD8$^+$ cytotoxic T cells ($P = 0.07$), a significant increase in CD4$^+$ T helper cells and Foxp3$^+$ regulatory T cells (Fig. 3E), but no change in PD-1 expression by any of these subpopulations. Thus, short and long-term Bcl-xL inhibition in vivo counteracts the ageing of TANs and increases the proportion of tumor-infiltrating T lymphocytes.

## Bcl-xL blockade reduces lung tumor growth

We next sought to evaluate the effect of Bcl-xL inhibition on tumor development when used as monotherapy. We treated KP tumor-bearing mice for an extended period of 3 weeks with A-1331852 and performed micro-computed tomography (μCT) imaging of the lungs before treatment initiation and every week thereafter (see scheme Fig. 4A). Bcl-xL blockade significantly delayed tumor growth after 2 weeks of treatment (Fig. 4B,C). On average, tumors in the control group doubled their size after 2 weeks, whereas tumors in treated mice were on average 1.3 times bigger compared to their size before treatment. Importantly, the anti-tumor response was maintained 1 week later (Fig. 4B). Staining by immunohistochemistry revealed a reduced number of tumor cells positive for Ki67 in treated mice, aligning with the decreased growth measured by μCT (Fig. 4D). Furthermore, there was a reduced proportion of total TANs, as well as significantly decreased total TAN, SiglecF$^+$ TAN and PD-L1$^+$ TAN numbers in treated compared to control tumors (Fig. 4E).

Because long-term A-1331852 reduced the total number of TANs, we wondered if Bcl-xL blockade modulates the circulating neutrophil pool. We observed that A-1331852 induced a significant increase in blood neutrophils (Fig. 4F). In healthy animals, a short 3-day treatment also augmented blood neutrophil abundance (Fig. 4G). This suggests that Bcl-xL blockade positively impacts on granulopoiesis or neutrophil egress from the bone marrow into the circulation. As the 3-week treatment had a significant impact on all TANs, we tested whether selectivity toward SiglecF$^+$ cells could be restored by introducing a pause in treatment. Specifically, instead of daily administration of A-1331852, we opted for a 5-day ON/2-day OFF regimen for the same total duration (Fig. EV4A). Remarkably, intermittent treatment with A-1331852 did not alter total TAN abundance; SiglecF$^-$ TANs became the major subset with almost 80% of total TANs, while SiglecF$^+$ TANs were very significantly reduced (Fig. EV4B).

Although our results demonstrate that ageing TANs are particularly sensitive to Bcl-xL blockade, the tumor growth delay could be due to the action of A-1331852 on other cell types, including tumor cells. In human NSCLC cell lines, Navitoclax and

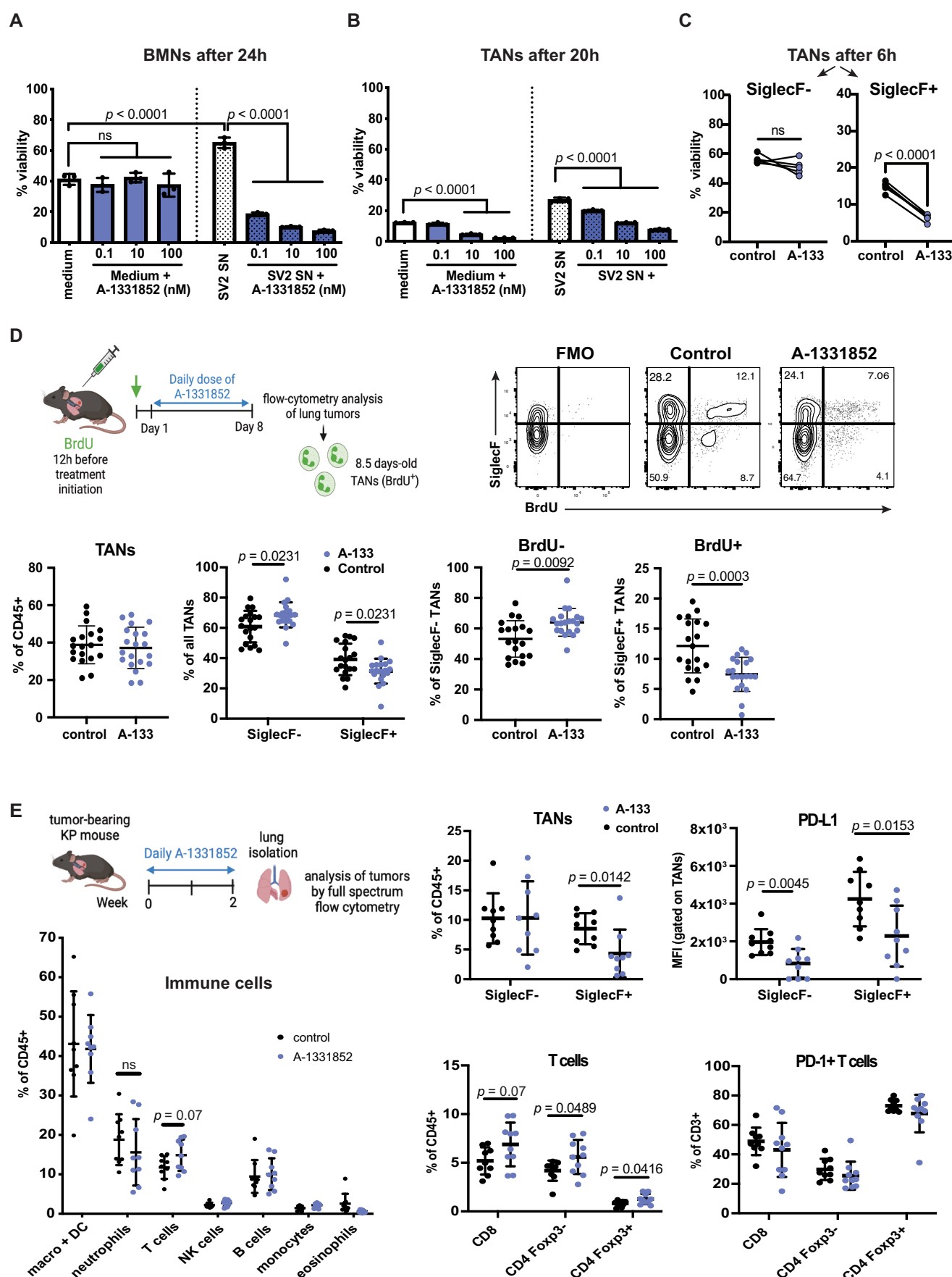

**Figure 3. A-1331852 reduces TAN survival in vitro and in vivo.**

(A, B) Bone marrow (A) and TAN viability (B) (% of live cells out of total) after 24 h in vitro culture in medium only or with increasing doses of A-1331852. (C) SiglecF⁻ and SiglecF⁺ TAN viability (% of live cells out of total) after 6 h in vitro culture in medium only or with 50 nM of A-1331852. TANs were obtained from $n = 5$ tumors. (D) Prevalence of total, SiglecF⁺, SiglecF⁻, BrdU⁻SiglecF⁻ and BrdU⁺SiglecF⁺ TANs. Each data point represents a single tumor. For control group, $n = 18$ tumors from three mice and $n = 20$ tumors from three A-1331852-treated mice. (E) Percentage of the different immune cell populations in KP tumors in mice treated for 2 weeks with A-1331852. Each data point corresponds to one tumor. $n = 8$ tumors from two control mice were analyzed and $n = 9$ tumors from three A-1331852-treated mice. Data information: All data are shown as mean ± SD. For (A, B), significance was determined by ordinary one-way ANOVA with Tukey's multiple comparisons test. For (A), the experiment was replicated three times. For (C–E), significance was based on a two-tailed Student's $t$ test (paired analysis for (C)). ns non-significant. Source data are available online for this figure.

A-1331852 were not cytotoxic as single agents but augmented the cell death response to chemotherapies (Tan et al, 2011; Potter et al, 2021). However, A-1331852 was reported to reduce tumor growth in a subcutaneous xenograft model of NSCLC (Leverson et al, 2015). To evaluate if A-1331852 affects tumor cell proliferation or viability, we used two $Kras^{Lox-STOP-Lox-G12D/WT}$; $p53^{Flox/Flox}$ tumor-derived cell lines, SV2 and T5, and incubated them with increasing concentrations of the compound. Cell viability measurements after 48 and 72 h demonstrated that the cells were resistant even to high doses (Fig. EV5A). Moreover, in a clonogenic assay performed on SV2 cells, the number of colonies did not vary at any drug concentration used compared to controls (Fig. EV5B). Finally, ex vivo administration of A-1331852 did not significantly reduce the viability of cells obtained from the Ly6G negative tumor fraction, which are mainly composed of tumor cells but also other immune cells and stromal cells (Fig. EV5C). These data collectively favor the hypothesis that Bcl-xL blockade, at least when used as a single agent, does not directly impact lung tumor epithelial cells.

To assess if part of the effects of Bcl-xL blockade on tumor progression are nevertheless neutrophil-independent, we initiated a regimen of neutrophil depletion using daily anti-Ly6G + anti-rat antibodies (Boivin et al, 2020) 2 days before A-1331852 treatment initiation and continued the combination for 2 weeks. Neutrophil depletion alone showed a trend toward decreased tumor growth as analyzed by µCT (Fig. EV5D). While A-1331852-treated mice confirmed the tumor growth delay, this effect was partly lost upon combined A-1331852 and neutrophil depletion. Of note, anti-Ly6G + anti-rat antibodies effectively diminished the abundance of blood neutrophils over this 2-week period but only partially decreased that of TANs, suggesting that the importance of TANs in these conditions is underestimated. Together, these results indicate that the effect of A-1331852 on tumor growth is at least partly mediated through TAN targeting.

## G-CSF treatment augments the anti-tumor efficacy of Bcl-xL blockade

Cancer patients are often being administered G-CSF as a recombinant protein to counteract chemotherapy-induced neutropenia. We wanted to know if this clinical agent could be used in conjunction with TAN-targeting A-1331852, reasoning that (i) it could act against A-1331852 by increasing neutrophil production and their infiltration into tumors, augmenting the total pool of TANs, or (ii) it could potentiate the effects of A-1331852, by providing younger neutrophils while Bcl-xL blockade selectively removes older TANs, together shifting the balance toward young (and possibly anti-tumor) TANs (Fig. 5A). To discriminate between these two possibilities, we combined A-1331852 treatment with

G-CSF injections for 3 weeks and measured tumor volumes with µCT before treatment initiation and 3 weeks later (Fig. 5B). As reported above (see Fig. 4B), Bcl-xL blockade diminished tumor growth significantly. Although G-CSF alone showed only a trend toward decreased tumor growth, it accentuated the anti-tumor response of A-1331852, with 25% tumors (6 out of 24) that regressed after 3 weeks, compared to only 1 out of 26 tumors in the single A-1331852 treatment group (Fig. 5C). 48 h before sacrifice, we injected mice with BrdU and measured by flow cytometry the proportions of BrdU⁺ TANs, representing newly recruited cells. In control and A-1331852-treated mice, there were almost no BrdU⁺ neutrophils within the tumors, whereas they represented around 10% TANs when G-CSF was used, indicating that G-CSF catalyzed neutrophil accumulation in tumors and that A-1331852 did not prevent it (Fig. 5D).

Because TAN prevalence often correlates with decreased T-cell infiltration and lower efficacy of immunotherapies (Faget et al, 2021), we tested if A-1331852 would sensitize tumors to anti-PD-1 that is inefficient in this model (Pfirschke et al, 2016). Combination treatment revealed a trend toward increased infiltration by CD8 T cells but no improvement compared to A-1331852 alone (Appendix Fig. S2A–C). Accordingly, T cells did not re-localize from the tumor periphery to the tumor interior upon A-1331852 treatment (Appendix Fig. S2D). Therefore, Bcl-xL blockade does not sensitize tumors to anti-PD-1 in this experimental system but displays increased anti-tumor efficacy when used in combination with a clinical anti-neutropenic agent.

## TANs have a higher expression of Bcl-xL compared to tumor-adjacent neutrophils in human LUAD

To interrogate the value of our results in humans, we first visualized single-cell RNA sequencing data obtained from patients with NSCLC. *BCL2L1* expression was enriched in N5 (Fig. 6A), a TAN subset specifically associated with poor prognosis (Zilionis et al, 2019a). In comparison, *BCL-2* was almost not expressed in any of the subsets, while *BCL2A1* and *MCL1* were strongly expressed, especially in N4/N5 and in N1, respectively (Fig. 6A).

Next, to monitor Bcl-xL protein, we performed immunofluorescence staining of human LUAD tissue samples with anti-MPO to identify neutrophils and TANs, and with anti-Bcl-xL. By quantifying and comparing Bcl-xL expression levels in MPO-positive cells in tumors and tumor-adjacent non-cancerous tissue, we demonstrated that TANs express significantly more Bcl-xL compared to peritumoral neutrophils (Fig. 6B; Appendix Table S1).

Finally, human neutrophil survival was augmented after 24 h of incubation with a supernatant prepared from the A549 human lung tumor cell line (Fig. 6C), which was accompanied by significantly

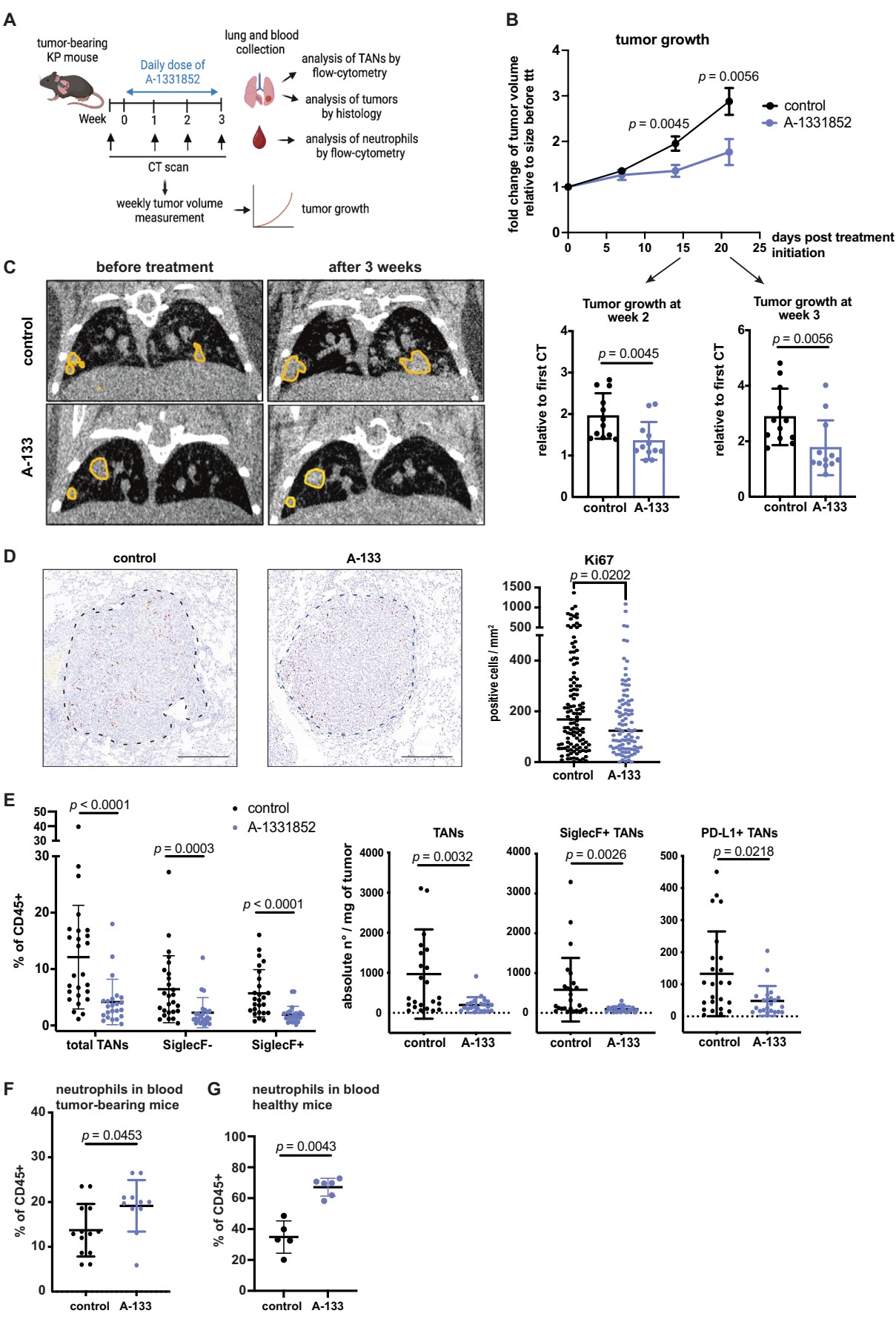

**Figure 4. Bcl-xL blockade reduces tumor growth in vivo.**

(A) Scheme showing the experimental design. Tumor-bearing mice were treated daily for 3 weeks. For these mice and for controls, tumors were measured one day before treatment initiation and then once weekly until the endpoint. (B) Evolution of tumor volume over 3 weeks in control mice or A-1331852-treated mice and dot plots showing the number of tumors analyzed ($n = 12$ tumors per group). Data show the ratio of tumor volumes relative to the initial volume size before treatment. (C) Examples of µCT scans of KP lungs with highlights of tumors in yellow before treatment initiation and 3 weeks post-treatment. (D) Representative IHC images and dot plot showing the quantification of Ki67$^+$ cells. Measurements are reported as the number of positive cells per mm$^2$ of lesion area. Each dot represents a single lesion analyzed. Scale bars: 200 µm. (E) Left, percentage of total TANs, SiglecF$^-$ and SiglecF$^+$ TANs out of CD45$^+$ cells in control ($n = 24$ tumors) and A-1331852-treated mice ($n = 22$ tumors analyzed). Right, absolute numbers are shown for total TANs, SiglecF$^+$ TANs, and PD-L1$^+$ TANs per tumor. (F) Percentage of neutrophils in blood from tumor-bearing mice in controls ($n = 13$) or after 3-weeks treatment ($n = 11$). (G) Percentage of neutrophils in blood from healthy mice in controls ($n = 5$) or after 3-days treatment ($n = 6$). Data information: Data are shown as mean ± SD, except for (D), where the median is shown, and except for (B), where data are shown as mean ± SEM. Significance was determined by the Mann–Whitney test for (B, E, F, G) and unpaired $t$ test for (D). Source data are available online for this figure.

induced Bcl-xL (Fig. 6D). More importantly, A-1331852-mediated Bcl-xL blockade efficiently repressed the supernatant-induced neutrophil survival, without interfering with basal neutrophil survival (Fig. 6E). Altogether, these findings support our mouse data and provide a path forward to consider Bcl-xL-dependent TAN targeting clinically.

## Discussion

The importance of the tumor microenvironment (TME) in cancer progression is widely recognized (Hanahan, 2022). Processes such as matrix remodeling, new vessel formation and immune cell modulation contribute to tumor cell survival and proliferation. Immune cells need to adapt to the challenges imposed by the TME, such as insufficient nutrients and oxygen. These conditions eventually impact their survival and functionality and are often linked to the acquisition of pro-tumorigenic properties (Strauss et al, 2021; Ramel et al, 2021). For instance, positron emission tomography imaging has shown that myeloid cells in tumors consume high amounts of glucose (Reinfeld et al, 2021), and our previous research demonstrated that TANs rely on increased glucose metabolism to survive in the TME and differentiate into tumor-promoting SiglecF$^+$ cells (Ancey et al, 2021a). The prolonged survival of TANs may be a common characteristic in LUAD and other solid malignancies. In a mouse model of lung squamous cell carcinoma, for example, the presence of SiglecF$^+$ TANs (Mollaoglu et al, 2018) suggests an increased lifespan, which remains to be established. Similarly, TANs in human hepatocellular carcinoma (HCC) exhibit enhanced survival ex vivo, attributed to increased autophagy (Li et al, 2015).

In addition to metabolic adaptations, the survival and behavior of TANs are influenced by a variety of molecules secreted within the TME (Duits and de Visser, 2021). While assessing the modulation of neutrophil survival by tumor-secreted factors in vivo is difficult, several ex vivo studies demonstrated that it is stimulated upon incubation with supernatants of various tumor cell lines, including head and neck cancer (Wu et al, 2011; Dumitru et al, 2012). In the present research, we show that lung tumor-secreted GM-CSF enhances neutrophil survival. GM-CSF, which was reported to extend neutrophil survival in non-cancer contexts (Colotta et al, 1992; Lee et al, 1993), was also shown to regulate the functionality of neutrophils in different mouse models of cancer. For example, tumor-derived GM-CSF activates neutrophils and enhances PD-L1 expression via JAK-STAT3 in gastric cancer (Wang et al, 2017), while the same signaling pathway endows neutrophils with

immunosuppressive activity in metastatic head and neck cancer (Pylaeva et al, 2022). In addition, breast tumor cells stimulate arginase-1 expression in myeloid cells through GM-CSF secretion, thus creating an immunosuppressive microenvironment (Su et al, 2021). Oncogenic KrasG12D was shown to stimulate the production of GM-CSF by pancreatic ductal cells, which led to the recruitment of CD8 T-cell-suppressing myeloid cells in vivo and subsequent cancer outgrowth (Pylaeva-Gupta et al, 2012). Interestingly, SiglecF$^+$ neutrophils were shown to be induced by GM-CSF in a mouse model of renal fibrosis (Ryu et al, 2022). Altogether, these studies and our findings position GM-CSF as a critical factor in promoting the survival of neutrophils and their phenotypic conversion toward tumor-supportive cells.

Beyond the specific role GM-CSF has on neutrophils as described in the present work, this growth factor has been shown to modulate cancer progression in opposite directions. CTLA-4 blockade-based anti-tumor immunity was augmented upon prior vaccination with autologous and irradiated tumor cells engineered to secrete GM-CSF, in patients suffering from metastatic melanoma or ovarian carcinoma (Hodi et al, 2008). However, in a mouse model of pancreatic ductal adenocarcinoma, the production of GM-CSF by tumor cells triggered the proliferation of c-Kit$^+$Lineage$^{neg}$ splenocytes and their differentiation into immunosuppressive myeloid cells, which were then recruited to the tumors and enhanced their growth (Bayne et al, 2012). In our study, we identify that GM-CSF acts directly and locally in the tumor microenvironment to sustain TAN survival.

While BH3 mimetics were developed to trigger tumor cell death, leading to the clinical approval of Venetoclax against several hematological malignancies (Lasica and Anderson, 2021), our data suggest their repurposing against non-malignant, tumor-supportive innate cells. Of note, Navitoclax and A-1331852 were shown to prime NSCLC cell lines for apoptosis when combined with chemotherapeutic agents (Kim et al, 2017; Potter et al, 2021). Thus, combined administration of a Bcl-xL inhibitor and chemotherapy could counteract disease progression through direct tumor cell killing and indirect, TAN-dependent targeting. In our mouse model, known to be refractory to immune checkpoint blockade (Pfirschke et al, 2016), we failed to sensitize tumors to anti-PD-1 upon Bcl-xL blockade. One possible explanation is that T cells, which reside mainly at the tumor periphery and often in tertiary lymphoid-like structures in the KP model (DuPage et al, 2011; Joshi et al, 2015; Faget et al, 2017), did not infiltrate the tumor mass upon Bcl-xL blockade. Thus, additional stimuli should be employed to trigger tumor homing of effector T cells. The utilization of BH3 mimetics in cancer patients suffers limitations from secondary effects. While Navitoclax has shown promise as

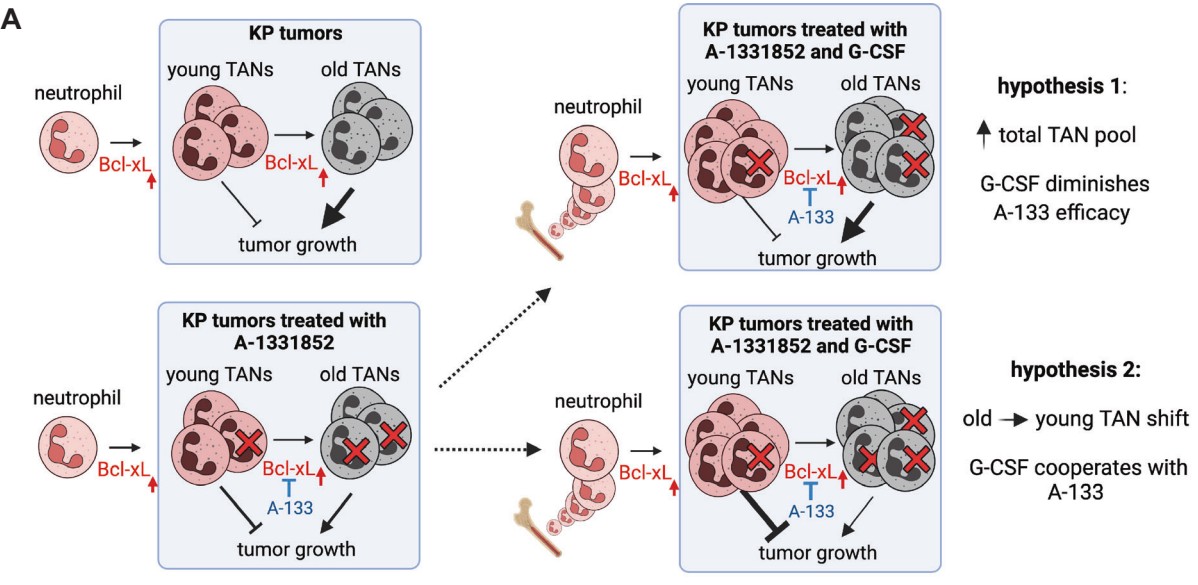

**A**

KP tumors
young TANs | old TANs
neutrophil → Bcl-xL↑ → Bcl-xL↑
tumor growth

KP tumors treated with A-1331852
young TANs | old TANs
neutrophil → Bcl-xL↑ → Bcl-xL↑ (A-133)
tumor growth

KP tumors treated with A-1331852 and G-CSF
young TANs | old TANs
neutrophil → Bcl-xL↑ → Bcl-xL↑ (A-133)
tumor growth

**hypothesis 1:**
↑ total TAN pool
G-CSF diminishes A-133 efficacy

KP tumors treated with A-1331852 and G-CSF
young TANs | old TANs
neutrophil → Bcl-xL↑ → Bcl-xL↑ (A-133)
tumor growth

**hypothesis 2:**
old → young TAN shift
G-CSF cooperates with A-133

**B**

tumor-bearing KP mouse
Daily A-1331852 +/- G-CSF
lung isolation → analysis of TANs by flow-cytometry

Week 0 1 2 3

↑ CT scan: follow-up of tumor growth

BrdU injection 48h before sacrifice

BrdU+ TANs: newly recruited neutrophils

**C**

**tumor growth after 3 weeks**

tumor size relative to first CT

| | control | A-133 | G-CSF | A + G |
|---|---|---|---|---|
| proportion of regressors | 0/24 | 1/26 | 0/27 | 6/24 |

p = 0.0002
ns
p = 0.0011

**D**

**BrdU+SiglecF-**

% of TANs

p < 0.0001
p = 0.0006
ns
ns

control   A-133   G-CSF   A + G

SiglecF / BrdU

| Control | A-133 | G-CSF | A + G |
|---|---|---|---|
| 32.1  0.11 | 25.1  0.35 | 35.2  0.62 | 31.3  0.52 |
| 67.3  0.48 | 73.8  0.73 | 56.6  7.61 | 54.8  13.4 |

◄ **Figure 5.   G-CSF potentiates the anti-tumor effect of Bcl-xL blockade.**

(A) Scheme representing the rationale and possible outcomes for combining Bcl-xL inhibition with G-CSF-mediated young neutrophil recruitment to tumors. (B) Scheme representing the treatment strategy. Control, A-1331852-treated, G-CSF, G-CSF + A-1331852-treated mice were treated daily, and tumor growth was measured by µCT once weekly for 3 weeks. $n = 3$ mice/group. BrdU injection 48 h before sacrifice enabled to track neutrophils that are newly recruited to the tumors. (C) Tumor growth was measured after 3 weeks, relative to the tumor volume size measured before treatment. Each data point represents a single tumor and the proportion of regressing tumors is shown below each group. $n = 24$ tumors analyzed for control and A-1331852 + G-CSF (A + G) groups, $n = 26$ for A-1331852 (A-133) and $n = 27$ for G-CSF-treated mice. (D) The percentage of BrdU+ newly recruited TANs (gated on CD11b+ Ly6G+ SiglecF− cells) and representative FACS plots showing the recruitment of BrdU+ neutrophils to the tumors in the indicated treatment conditions. $n = 10$ tumors analyzed for control, A-1331852 and G-CSF and $n = 8$ tumors for A-1331852 + G-CSF. Data information: Data are shown as mean ± SD. Significance was determined by the Kruskal–Wallis test followed by Dunn's multiple comparisons test. ns non-significant. Source data are available online for this figure.

potential cancer therapy for solid tumors (Tse et al, 2008; Leverson et al, 2015), its clinical application is limited, in part due to induced neutropenia when combined with chemotherapies (Leverson et al, 2015). Comparatively, the use of Bcl-xL-selective inhibitors in rats did not result in neutropenia and did not suppress human ex vivo granulopoiesis (Leverson et al, 2015). Our results align with these findings, as none of the mice treated with A-1331852 became neutropenic, even after 3 weeks of treatment. One anticipated issue is, however, the on-target toxicity Bcl-xL blockade has on platelets (Debrincat et al, 2015), which was observed with the use of Navitoclax in the clinics (Ploumaki et al, 2023). Since then, the development of proteolysis-targeting chimeras (PROTACs) to prevent the uptake of Bcl-xL-specific mimetics by thrombocytes effectively attenuated platelet depletion (Khan et al, 2019; Negi and Voisin-Chiret, 2022). Because we demonstrated that small amounts of A-1331852 are sufficient to inhibit tumor cell-enhanced human neutrophil survival, low-dose treatments or the use of PROTACs could provide beneficial effects for the patients with reduced or clinically manageable thrombocytopenia.

There is accumulating evidence for tumor-promoting roles of neutrophils in cancer. Nevertheless, neutrophils can also participate in tumor elimination. In early-stage human lung cancer, neutrophils mediate the tumor-killing capacities of T cells (Eruslanov et al, 2014), and neutrophils from healthy blood donors appear to exhibit tumor cell-killing activity in vitro (Yan et al, 2014). Recently, a neutrophil response has been described for successful tumor control upon immunotherapy in lung, colon (Gungabeesoon et al, 2023) and melanoma tumor mouse models (Hirschhorn et al, 2023). Because patients under chemotherapy as frontline treatment are at risk of developing neutropenia (Blayney and Schwartzberg, 2022), G-CSF is often administered to foster neutrophil production and their egress from the bone marrow (Mehta et al, 2015), (Lambertini et al, 2015). Concurrent administration of G-CSF with radiotherapy was shown to enhance neutrophil tumor-killing ability in mice (Takeshima et al, 2016). Our results with combined G-CSF administration and Bcl-xL blockade suggest a qualitative change of the tumor-infiltrating neutrophil pool, whereby old, tumor-supportive TANs are replaced by younger, possibly tumor-antagonizing neutrophils. Thus, understanding and targeting specific pathways of TAN ageing could become an interesting option to disrupt tumor support and favor the anti-tumor activity of these cells.

Neutrophil targeting is difficult to accomplish in vivo, but the plasticity of these cells in cancer progression may hold the key to more precise and specific targeting. Our present work proposes a safe approach to selectively target pro-tumoral TANs while preserving the healthy pool of neutrophils, which can be harnessed for their capacities to fight cancer.

# Methods

## Mouse model

Kras$^{\text{Frt-STOP-Frt-G12D/WT}}$;Tp53$^{\text{Frt/Frt}}$ (KP) mice, which were generated by crossing Kras$^{\text{Frt-STOP-Frt-G12D/WT}}$ (RRID:IMSR_JAX:008653) (Young et al, 2011) and Tp53$^{\text{Frt/Frt}}$ (RRID:IMSR_JAX:017767) (Lee et al, 2012) mice, were generously provided by D.G. Kirsch from Duke University Medical Center. These mice were bred in a mixed 129-C57BL/6 background. Mice were kept at 2–5 per cage and provided normal diet.

## Tumor initiation and follow-up

To activate the oncogenic KrasG12D and delete Tp53, tumors were initiated in 12–14-week-old mice through the intratracheal administration of $10^7$ plaque-forming units (PFU) of a commercially available adenoviral CMV-Flp vector (Ad5CMVFlpo, obtained from University of Iowa Viral Vector Core Facility) per mouse, to activate the oncogenic $Kras^{\text{G12D}}$ and induce the deletion of $Tp53$. To monitor the progression of tumors, mice were anesthetized using isoflurane and kept anesthetized throughout the scanning process. Lung images were captured using an X-Ray microtomography machine (µCT) (Quantum FX; PerkinElmer) with a voxel size of 50 µm, using retrospective respiratory gating. The volume of individual tumors was obtained using Osirix MD (Pixmeo, RRID:SCR_013618) and following the protocol described in bio-protocol.org/prep390.

## Mouse treatments

Both genders were used for the experiments. Treatment with A-1331852 was initiated at 16 weeks after tumor initiation. Treatment exclusion criteria were based on the health score sheets. The Bcl-xL inhibitor A-1331852 (Cat#: HY-19741, MedChemExpress) was given at a dose of 25 mg/kg. The dual Bcl-2 and Bcl-xL inhibitor Navitoclax (ABT-263, Cat#: HY-10087, MedChemExpress) was given at a dose of 100 mg/kg and The Bcl-2-specific inhibitor Venetoclax (ABT-199, Cat#: HY-15531 MedChemExpress) at a dose of 50 mg/kg. All the compounds for in vivo treatments were formulated in 10% DMSO, 40% PEG-300, 5% Tween Tween-80 and 45% saline and were administered daily orally.

For the BrdU assay, 2 mg of freshly prepared BrdU (Merck, 10280879001) was prepared into 100 µL of PBS and injected intraperitoneally. For immunotherapy treatment, anti-mouse PD-1 (clone 29 F.1A12, Bio X Cell, BE0273, RRID:AB_2687796) was intraperitoneally injected at a dose of 200 µg/mouse three times a

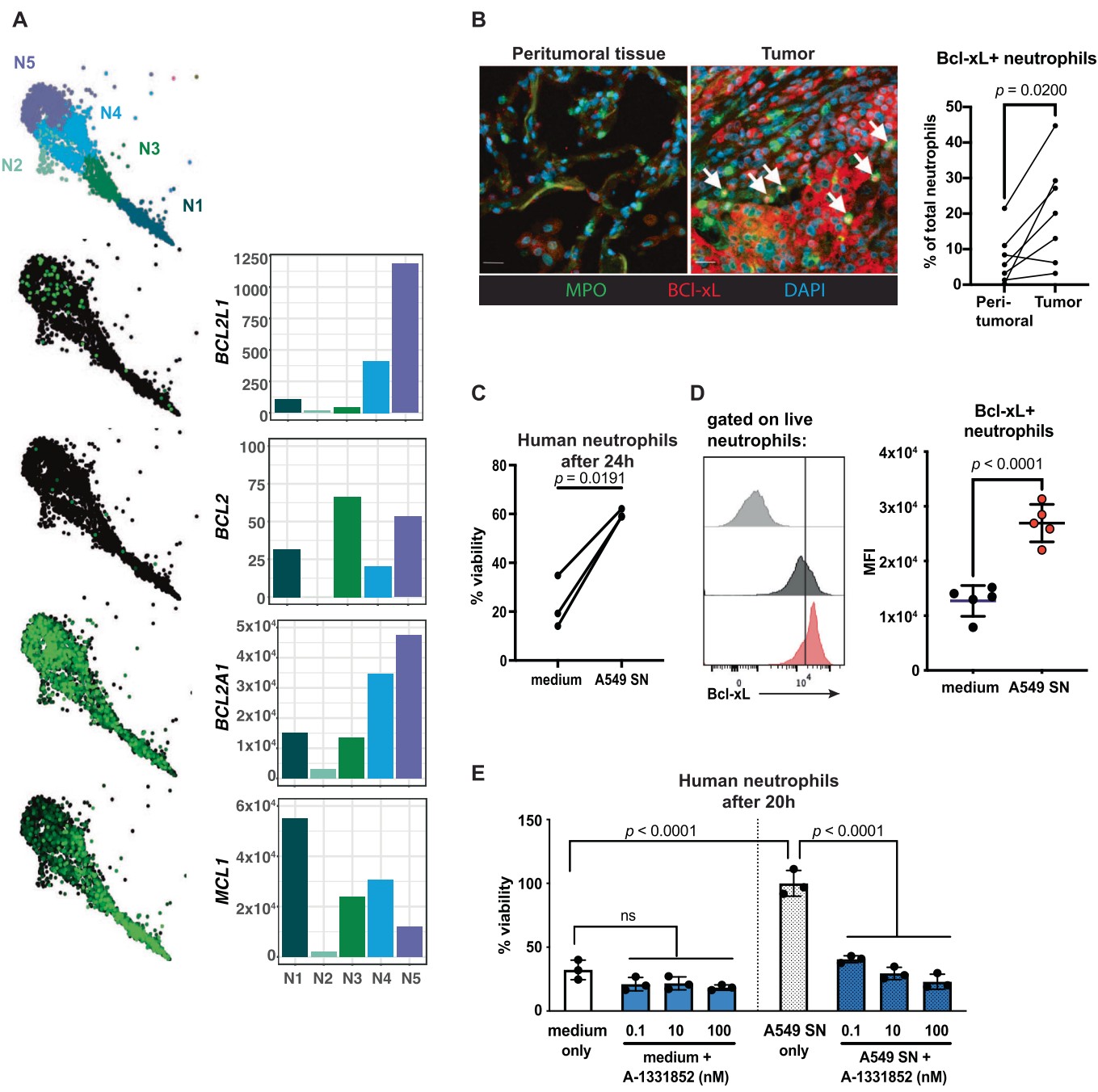

**Figure 6. Human neutrophils upregulate Bcl-xL in tumors and survive longer with tumor cell-conditioned medium.**

(A) Representative images of the available single-cell transcriptomics showing the five TAN subsets from patients with lung cancer. *BCL2L1*, *BCL-2*, *BCL2A1*, and *MCL1* expression are shown per subset. (B) Representative example of neutrophils (MPO⁺ cells) and Bcl-xL co-immunostaining in seven LUAD patient samples and quantification of the percentage of Bcl-xL⁺ neutrophils in matched tumor and peritumoral tissue. White arrows indicate MPO⁺ Bcl-xL⁺ cells. Scale bars: 20 μm. (C) Percentage of live healthy blood donor-derived neutrophils after 24 h of incubation with A549 tumor cell line supernatant. (D) Representative flow cytometry and quantification showing Bcl-xL upregulation in human blood-derived neutrophils upon incubation with tumor cell line supernatant after 24 h. $n = 5$ donors. (E) Viability of human neutrophils with A549 SN and increasing doses of A-1331852 in vitro after 18 h of incubation. $n = 3$ biological replicates. Data information: For (D, E), data are shown as mean ± SD. For (B–D), significance was determined with a paired *t* test. For (E), significance was determined with a one-way ANOVA followed by Šídák's multiple comparisons test. ns non-significant. Source data are available online for this figure.

week for 2 weeks. For G-CSF treatment, recombinant mouse G-CSF (PeproTech, 250-05) was intraperitoneally injected daily at a dose of 10 µg for 3 weeks. Neutrophil depletion was performed using a double antibody approach with anti-Ly6G antibody (clone 1A8, #BP0075-1) injection followed by anti-rat Kappa immunoglobulin (clone MAR 18.5, #BE0122), according to a published method (Boivin et al, 2020). The antibodies and corresponding isotype controls (#BP0290 and #BP0089) were purchased from BioXCell.

## Immunohistochemistry and immunofluorescence on KP mouse slides

After dissection, lungs with tumors were isolated, the lobes were separated, fixed with 3.7% formaldehyde solution (Sigma) overnight, and then paraffin-embedded. Blocks of tissues were cut into sections of 4 µm of thickness. For immunohistochemistry (IHC), slides were dewaxed and antigen retrieval was performed using 10 mM sodium citrate. Blocking was performed with 5% goat serum. The slides were then stained with primary antibodies overnight at 4 °C, washed and incubated with secondary antibodies for 1 h at room temperature. Antibody used for IHC is anti-Ki67 (1:100, ThermoFisher Scientific, MA5-14520). After washing, slides were incubated for 40 min with anti-rabbit Immpress horseradish peroxidase (HRP, Vector Laboratories, RRID:AB_2336529). The positive cells were then revealed using 3-3'-diaminobenzidine (DAB) substrate. Harris hematoxylin counterstain of the nuclei was subsequently performed.

For immunofluorescence, after dewaxing and antigen retrieval, slides were blocked with 5% goat serum and incubated with the following primary antibodies: anti-Bcl-xL (1:100, Abcam, ab32370) and anti-MPO (1:400, R&D SYSTEMS, Cat#: AF3667). Secondary antibodies are anti-rabbit Alexa 488 (1:500, ThermoFisher Scientific, A-21206) and anti-goat Alexa 568 (1:500, ThermoFisher Scientific, A-11057). For both IHC and immunofluorescence, tissue slides were scanned with the VS120-SL Olympus slide scanner at ×20 magnification, and quantification of positive cells on IHCs was performed using the QuPath software.

To assess the infiltration of different T-cell subpopulations in tumors, multiplexing (4plex) immunofluorescence was performed by the EPFL histology core facility using the fully automated Ventana Discovery ULTRA (Roche Diagnostics) and with the Ventana solutions. Briefly, paraffin sections were dewaxed, rehydrated and incubated sequentially with the following antibodies: anti-pan-cytokeratin (1:100, Novusbio, NBP600-579), anti-CD8 (1:100, Dako, M7103), anti-CD4 (1:100, Invitrogen Cat#: 14-0042-82) and anti-Foxp3 (1:50, Invitrogen Cat#: 14-5773). The slides were incubated with primary antibodies for one hour at 37 °C and then with a secondary antibody anti-rabbit ImmPRESS™ HRP (1:200, Vector Laboratories, Cat#: MP-7401). They were then sequentially revealed with the following TSA kits: the FAM (Roche Diagnostics, 07988150001), Red 610 (Roche Diagnostics, 07988176001), Rhodamine-6G (Roche Diagnostics, 07988168001) and Cyanine 5 (Roche Diagnostics, 07551215001) and counterstained with DAPI for nuclear staining.

## Tumor and cell line-derived supernatant production

KP tumors were dissociated into single-cell suspensions as described previously (Faget et al, 2017). Briefly, individual tumors

were digested by mechanical dissociation using the GentleMACS tissue dissociator (Miltenyi) coupled with enzymatic digestion using DNase I (0.02 mg/ml) and collagenase (1 mg/mL), resuspended in DMEM without FBS. Single-cell suspensions were then resuspended in complete DMEM (10% FBS and 1% PenStrep) and were plated at a density of $10^6$ cells/mL. After 24 h of incubation, the supernatant (SN) was collected and passed through 0.22-µm filters. The supernatants were kept at −80 °C until use. For SN produced from SV2 cells, $10^6$ cells/mL were cultured in DMEM, and SN was retrieved and filtered after 24 h.

## GM-CSF measurement

GM-CSF presence in the supernatant derived from digested tumors was measured with the enzyme-linked immunosorbent assay ELISA MAX™ Standard Set Mouse GM-CSF (432201, Biolegend), following the manufacturer's protocol.

## Flow cytometry

Single-cell suspensions obtained from tumors with mechanical and enzymatic digestion were resuspended in FACS buffer (2%FCS and 2 mmol/L EDTA) with FcR-block (anti-CD16/32, BioLegend) and stained with antibodies (Appendix Table S2), in darkness and for 20 min on ice. For intracellular staining of Bcl-xL, cells were fixed in BD Cytofix/Cytoperm (BD #554714) for 30 min on ice, then washed and incubated with Permeabilization buffer (Invitrogen #00–8333-56). For quantitative assessment of neutrophils in tumors, 15 µL of counting beads (CountBright™, Invitrogen, Cat#: C36950) were added to the sample before acquisition. For the experiment in Fig. 3E, acquisitions were performed with the full-spectrum analyzer Cytek Aurora (Cytek Biosciences). For the experiments, acquisitions were performed using the LSRII SORP (Becton Dickinson), a 5-laser and 18-detector analyzer at the EPFL Flow Cytometry Core Facility. Data analysis was performed using FlowJo (FlowJo LLC ©). Flow cytometry gating strategy is indicated in Appendix Fig. S1.

## Cell lines

The human cell line A549 (RRID:CVCL_0023) was purchased from ATCC and cultured in RPMI medium supplemented with 10% FBS. The murine cell lines SV2 and T5 were generated in our laboratory from a single $Kras^{Lox-STOP-Lox-G12D/WT}$; $p53^{Flox/Flox}$ and a $Kras^{Frt-STOP-Frt-G12D/WT}$; $p53^{Frt/Frt}$ lung tumor, from a male and a female mouse, respectively. Briefly, tumors were digested into single-cell suspensions and cultured for 25 passages before experimentation in DMEM supplemented with 10% FBS and 1% PenStrep. Cells were cultured at 37 °C with 5% $CO_2$. Mycoplasma tests were performed to ensure mycoplasma-free cell cultures.

## Tumor cell viability assay and clonogenic assay

SV2 lung tumor cells were seeded at 3000 cells per well in 96-well plates and treated with A-1331852 diluted in tenfold steps (range) and cell viability was measured using PrestoBlue (ThermoFisher Scientific, Cat#: A13261) after 48 h and 72 h of incubation. To test the proliferative capacity of tumor cells, a clonogenic assay was performed by incubating 100, 200, or 400 single SV2 cells into

six-well plates with increasing doses of A-1331852 and letting them grow for 1 week. Incubation with crystal violet solution enabled to stain for the colonies formed, which were then counted manually.

## Mouse neutrophil isolation and survival assay

Neutrophils from different tissues were isolated using anti-Ly6G MicroBeads UltraPure magnetic beads (clone REA526, Miltenyi Biotec, 130-120-337) according to the manufacturer's instructions. For bone marrow neutrophils, bone marrow was collected from femurs and tibia by flushing of the bone with PBS and filtered through a 40-µm filter. For healthy lung neutrophils, lungs from healthy mice were prepared the same way as lung tumors. For the survival assays, freshly isolated bone marrow neutrophils ($10^5$ cells) were incubated in 96-well plates and in 200 µL of tumor or SV2 cell-derived SN for 24 h. Neutralization of GM-CSF was done by pre-incubation of the SN with 10 ng/mL of anti-GM-CSF (PreproTech, Cat#500-P65) before adding it to the neutrophils. For the survival assay with GM-CSF, neutrophils were incubated with 10 ng/mL of recombind murine GM-CSF (PreproTech, Cat#315-03). For the experiments related to pathways inhibitors, Stattic (STAT3 inhibitor, MedChemExpress, Cat#: HY-13818), Ruxolitinib (JAK1/2 inhibitor, MedChemExpress, Cat#: HY-50856), MLN120B (IKKß inhibitor, MedChemExpress, Cat#: HY-15473) and LY294002 (PI3K inhibitor, MedChemExpress, Cat#: HY-10108) were resuspended in DMSO and diluted at the concentrations indicated in the figure legends. TANs were isolated from single-cell tumor suspensions using the anti-Ly6G MicroBeads UltraPure magnetic beads. Isolated TANs were stained with anti-SiglecF-Pe-Vio615 antibody (Cat#: 130-112-172) for 15 min on ice, before incubation, to specifically determine the effect of A-1331852 on SiglecF$^+$ and SiglecF$^-$ TANs. For viability determination of Ly6G- cells from KP tumors, TANs were isolated from single-cell tumor suspensions using anti-Ly6G positive selection with magnetic beads. The remaining Ly6G negative fraction was collected and incubated with 10 nM of A-1331852 for 18 h.

The viable cells were measured by flow cytometry using the LIVE/DEAD™ fixable blue dead stain kit (Invitrogen, Cat#: L23105) staining for dead cell exclusion.

## Detection of apoptotic neutrophils

Measurement of apoptotic neutrophils was done by double staining with Annexin V (FITC) and 7-AAD (detection kit, BioLegend, #640922) and analyzed by flow cytometry. TNF (Preprotech: Cat#: 315-01 A) was used at 5 ng/mL as a positive control for neutrophil apoptosis. Briefly, 100,000 neutrophils were resuspended in 100 µL of Annexin V binding buffer and 5 µL of Annexin V-FITC and 5 µL of 7-AAD were added. The samples were kept in the dark for 15 min at room temperature and analyzed by flow cytometry within one hour. Additionally, cleaved-caspase-3 (Asp175, Cell Signalling, rabbit mAb #9664) was detected by western blot.

## RNA isolation and real-time PCR

TRIzol Reagent (Invitrogen, 15596018) was utilized to extract the total RNA following the manufacturer's instructions. The High-Capacity cDNA Reverse Transcription Kit (ThermoFisher Scientific, 4368814) was employed to synthesize cDNAs from 1 µg total RNA. Real-time PCR was carried out using 5 ng of cDNA with the Taqman universal PCR master mix (ThermoFisher Scientific, 4324018) and the following Taqman probes: *Bcl2l1*: Mm00437783_m1; *Bcl2a1*: Mm03646861_mH; *Bcl-2*: Mm00477631_m1; *Mcl1*: Mm01257351_g1; *SiglecF*: Mm00523987_m1; *Ifit3*: Mm01704846_s1; *Ifnb1*: Mm00439552_s1; *Irf3*: Mm00516784_m1; *Dxd58*: Mm01216853_m1 and the gene expression level normalization was done with *Rpl30*: Mm01611464_g1 or with *Hprt*: Mm00446968_m1.

## Mouse mRNA sequencing and human TANs single-cell sequencing

Data from Fig. EV1A,B come from a previously published RNA sequencing (data ref: Ancey et al, 2021b). For Fig. 6A, filtered normalized counts for human TANs were retrieved from https://singlecell.broadinstitute.org/single_cell/study/SCP739/single-cell-transcriptomics-of-human-and-mouse-lung-cancers-reveals-conserved-myeloid-populations-across-individuals-and-species#study-summary (human_gene_expression_matrix.tsv). Counts were summed per patient and per tumor neutrophil population (N1-5) and the mean expression of *BCL2L1*, *BCL-2*, *BCL2A1* and *MCL1* is shown for each population.

## Human gene expression data

Five public transcriptome datasets (Appendix Table S3) have been combined to assess the overall survival of LUAD patients. To ensure data consistency and allow meaningful comparisons, background subtraction and normalization were performed on each dataset using the robust multi-array average (RMA) method from the affy package (Bolstad et al, 2003). In addition, to address potential batch effects that might arise due to differences in data collection and processing protocols, we employed the ComBat method from the sva package (Johnson et al, 2007). This method utilizes an empirical Bayes framework to adjust for known batch effects, harmonizing the combined dataset. The median expression of *CSF2* was then used to stratify patients into high or low groups and their survival probability was compared using a log-rank test.

## Western blot

Protein extracts from isolated neutrophils were obtained through lysis and sonication of the cells in RIPA buffer (20 mmol/L Tris pH8, 50 mmol/L NaCl, 0.5% sodium deoxycholate, 0.1% SDS, 1 mmol/L Na3VO4, protease inhibitor cocktail (Roche, 11836145001). Protein quantification was done using the Bradford assay. The following antibodies were used: anti-Bcl-xL (1:1000, Abcam, ab32370) and γ-tubulin (1:3000, Invitrogen, Cat#MA1-850).

## Immunofluorescence on human LUAD samples

Briefly, the formalin-fixed paraffin-embedded tissue sections were deparaffinized, rehydrated, and were subjected to antigen retrieval in EnVision FLEX Target Retrieval High pH Solution (Tris/EDTA, pH9.0; Agilent) at 95 °C for 20 min using PT Link (Agilent Technologies, Santa Clara, CA, USA). The sections were then permeabilized with 0.1% Triton X-100 in PBS for 15 min and blocked with 5% V/V donkey serum (Jackson ImmunoResearch, 017-000-121). The primary antibodies, anti-MPO (R&D SYSTEMS, AF3667) and anti-Bcl-xL (Abcam, ab32370), were added for overnight incubation at 4 °C. The primary antibodies were

**The paper explained**

**Problem**

Tumor-associated neutrophils (TAN) are emerging as an important innate immune cell type for cancer growth control and therapy response. In genetically engineered mouse models of lung adenocarcinoma, the lifespan of TANs is substantially augmented compared to normal neutrophils, which endows them with tumor support capability. However, the molecular mechanisms accounting for their aberrant ageing, and whether they could be exploited to disrupt tumor-supportive TANs and diminish cancer growth, are currently unknown.

**Results**

Here we demonstrate that the expression of the anti-apoptotic protein, Bcl-xL, raises in neutrophils homing to tumors, implicating GM-CSF and JAK/STAT signaling. In a Kras(G12D/WT); p53(Frt/Frt) model of lung cancer, blocking Bcl-xL activity impairs TAN ageing, thus diminishing the abundance of long-lived, tumor-supportive TANs, while preserving young TANs. Long-term treatment decreased tumor growth, which was further improved upon concomitant administration of G-CSF, a drug used to combat neutropenia in patients.

**Impact**

Small molecule Bcl-2 family inhibitors are used clinically to trigger tumor cell apoptosis. Here, an anti-Bcl-xL compound has been repurposed to counteract the ageing process of TANs and their tumor support. This approach has the advantage of selectively targeting pro-tumor TANs without affecting normal neutrophils and younger TANs that may carry anti-tumor function or be important for the success of therapy. This study therefore represents a proof-of-concept for the targeting of certain innate immune cells, which could complement existing therapies in the fight against cancer.

tumors enabled to randomize mice. Tumor volume measurements were performed blindly. The figure legends contain information on the statistical details of the experiments, including the number of repeats performed and the statistical tests used. The normality of distribution was tested using the Kolmogorov–Smirnov Test. Multiple comparisons of normally distributed samples were carried out using ANOVA, and Student $t$ tests were used for two independent samples. For nonparametrically distributed samples, the Kruskal–Wallis ANOVA with the Bonferroni correction for multiple comparisons, and Mann–Whitney $U$ test for two independent samples. Statistical analysis was performed using Prism 9 software.

## Study approval

All mouse experiments were performed with the permission of the Veterinary Authority of the Canton de Vaud, Switzerland, License number VD2931. The human blood samples used for fundamental research were anonymized prior to transmission by the Transfusion Interrégionale CRS, which waived the need for a cantonal ethics authorization. The human lung tumor sections were archived at the Institut Jules Bordet biobank and obtained with the agreement from the local ethics committee. The experiments on human material conformed to the principles set out in the WMA Declaration of Helsinki and the Department of Health and Human Services Belmont Report.

### Graphics

Figures 1A,E, 3D,E, 4A, 5A,B, EV3F, EV4A, Appendix Fig. S2A, and synopsis graphics were created with BioRender.com.

## Data availability

Microscopy and micro-computed tomography scan files are deposited in BioStudies, S-BIAD930. Flow cytometry files are deposited in FlowRepository. Source fcs files for Fig 1C: FR-FCM-Z7YP; Fig 1D: FR-FCM-Z7YQ; Fig 3D: FR-FCM-Z7YS; Fig 3E: FR-FCM-Z7YT; Fig 4G: FR-FCM-Z72Y; Fig 4E: FR-FCM-Z722; Fig 5D: FR-FCM-Z723; Fig 6D: FR-FCM-Z724.

## Peer review information

washed with washing solution 0.05% Tween-20 in PBS followed by incubation with secondary antibodies, Alexa Fluor 488-conjugated Donkey anti-Goat (Jackson ImmunoResearch, 705-545-003) and Cy3-conjugated Donkey anti-Rabbit (Jackson ImmunoResearch, 711-165-152) diluted in donkey serum, for 1 h at room temperature. The secondary antibodies were washed and counterstained with DAPI for nuclear inspection. The slides were scanned using Axioscan7 (Carl Zeiss SA, Oberkochen, Germany) and double-positive cells were analyzed using QuPath's (v 0.3.2) classifier.

## Human blood neutrophil isolation and survival assay

Peripheral blood neutrophils were isolated from whole-blood samples of healthy donors. Neutrophils were isolated from the blood using a Polymorphprep (Progen, Cat# 1895) gradient and the remaining red blood cells were lysed with ACK-buffer. The purity was assessed to be >95% with Quick-Fix staining of CytoSpinned samples. The viability of neutrophils incubated with A549 cell line supernatant was assessed by flow cytometry using the LIVE/DEAD™ fixable blue dead stain kit (Invitrogen, Cat#: L23105). The viability of human neutrophils upon treatment with A-1331852 was determined using PrestoBlue.

## Statistics

The sample size for mouse experiments was based on a previous study (Faget et al, 2017). Before treatments, µCT scans to visualize

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

## Acknowledgements

We thank MJ Pittet (University of Geneva, Switzerland) for critical reading of the manuscript. We thank DIAPath—CMMI and the EPFL SV Histology Core Facility for their technical assistance. The CMMI is supported by the European Regional Development Fund and the Walloon Region. We thank J Parchet-Piccand and R Doenlen for their assistance with the animal studies. This work was supported by the Swiss National Science Foundation [310030_179324], the Chercher et Trouver Foundation, the Fonds de la Recherche Scientifique-FNRS under Grants [MISU F.6003.22] and [TLV—7.4574.21], the ULB "Actions de Recherche Concertées" (ARC consolidation), the Fonds Jean Brachet and the Association Jules Bordet.

## Author contributions

**Anita Bodac**: Conceptualization; Data curation; Formal analysis; Validation; Investigation; Visualization; Methodology; Writing—original draft; Writing—review and editing. **Abdullah Mayet**: Data curation; Formal analysis; Investigation. **Sarika Rana**: Data curation; Formal analysis; Investigation. **Justine Pascual**: Investigation. **Amber D Bowler**: Investigation; Methodology. **Vincent Roh**: Resources; Visualization. **Nadine Fournier**: Resources; Visualization. **Ligia Craciun**: Resources; Data curation. **Pieter Demetter**: Resources; Validation. **Freddy Radtke**: Resources; Supervision; Writing—review and editing. **Etienne Meylan**: Conceptualization; Formal analysis; Supervision; Funding acquisition; Validation; Visualization; Writing—original draft; Project administration; Writing—review and editing.

## Disclosure and competing interests statement

FR is a co-founder of Cellestia. EM serves in the scientific advisory board of InhaTarget Therapeutics. The remaining authors declare no competing interests.

# Expanded View Figures

**Figure EV1.  Bcl-xL expression analysis in neutrophils and TANs.**

(A) Gene set enrichment analysis (GSEA) showing downregulation of the apoptosis pathway in TANs compared to HLNs. (B) Volcano plot showing differentially expressed genes (DE) in TANs versus HLNs. The anti-apoptotic genes *Bcl2l1*, *Bcl2a1b* and *Mcl1* are highlighted in dark blue. (C) Real-time PCR showing *Bcl-2, Bcl2l1, Bcl2a1* and *Mcl1* gene expression in TANs ($n = 11$ biological replicates) normalized to expression in HLNs ($n = 5$). *Rpl30* was used as a reference gene. (D) Immunofluorescence staining of neutrophils (MPO) and Bcl-xL in tumors of KP mice. Scale bar: 100 μm. (E) Representation of neutrophil subsets in naive and tumor-bearing mice from the available single-cell transcriptomics. *SiglecF*, *Bcl2l1*, *Bcl-2*, *Bcl2a1b* and *Mcl1* expressions are highlighted in green. (F) Representative flow cytometry gating strategy of alive bone marrow-extracted neutrophils after 24 h incubation with medium or SV2 SN. Data information: For (A), statistical significance was calculated by permutation tests (number of random permutations $= 10^5$). For (B), differential gene expression was computed with limma and significance assessed with the moderated *t* test. Genes with *P* value < 0.01 are highlighted in red ($n = 1335$, $n = 471$ with LFC > 0, $n = 864$ with LFC < 0). Total number of genes tested $n = 5397$. (C) Data are shown as mean ± SD and significance was obtained with two-way ANOVA with Sidàk's multiple comparisons test. ns non-significant.

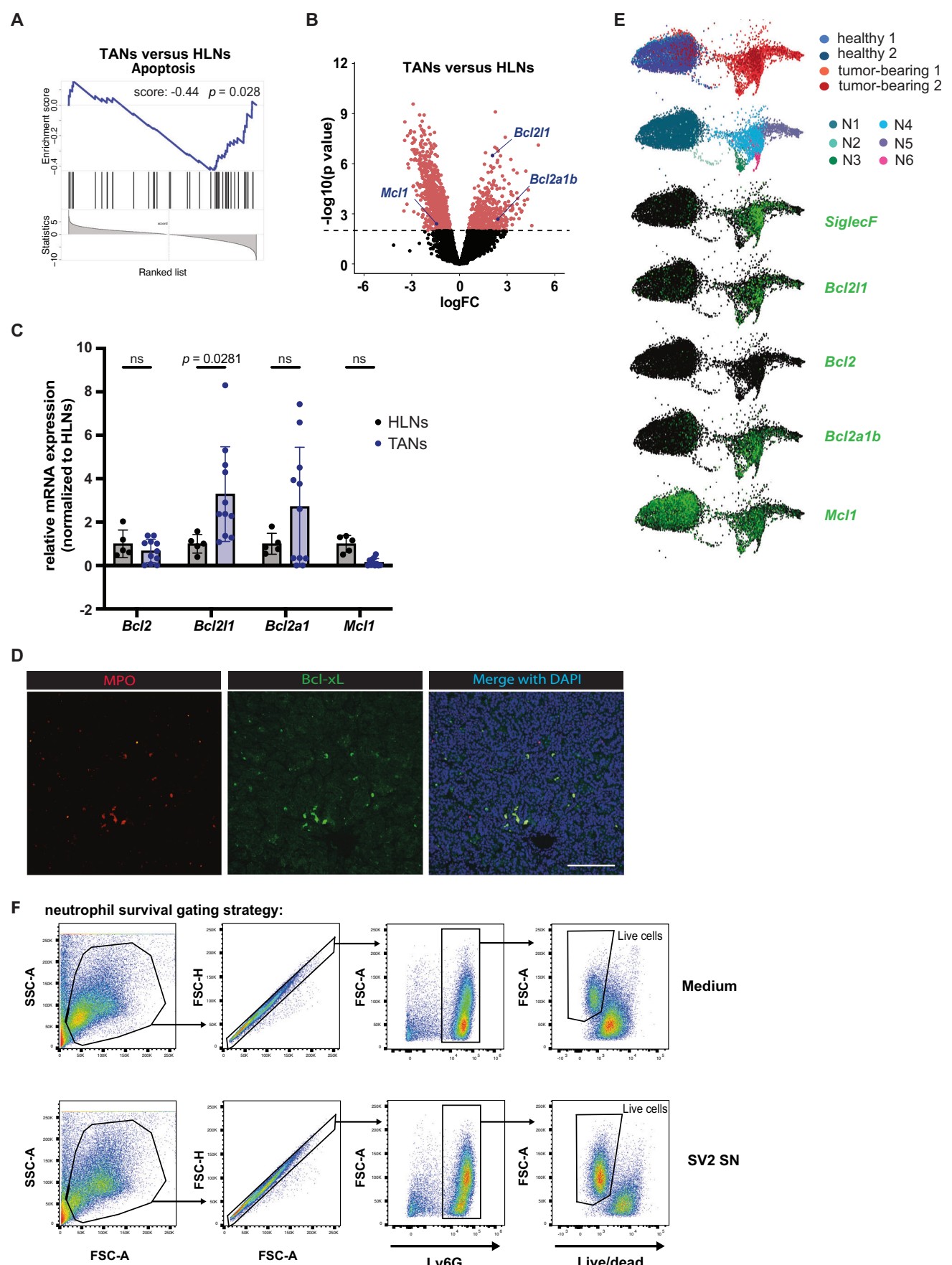

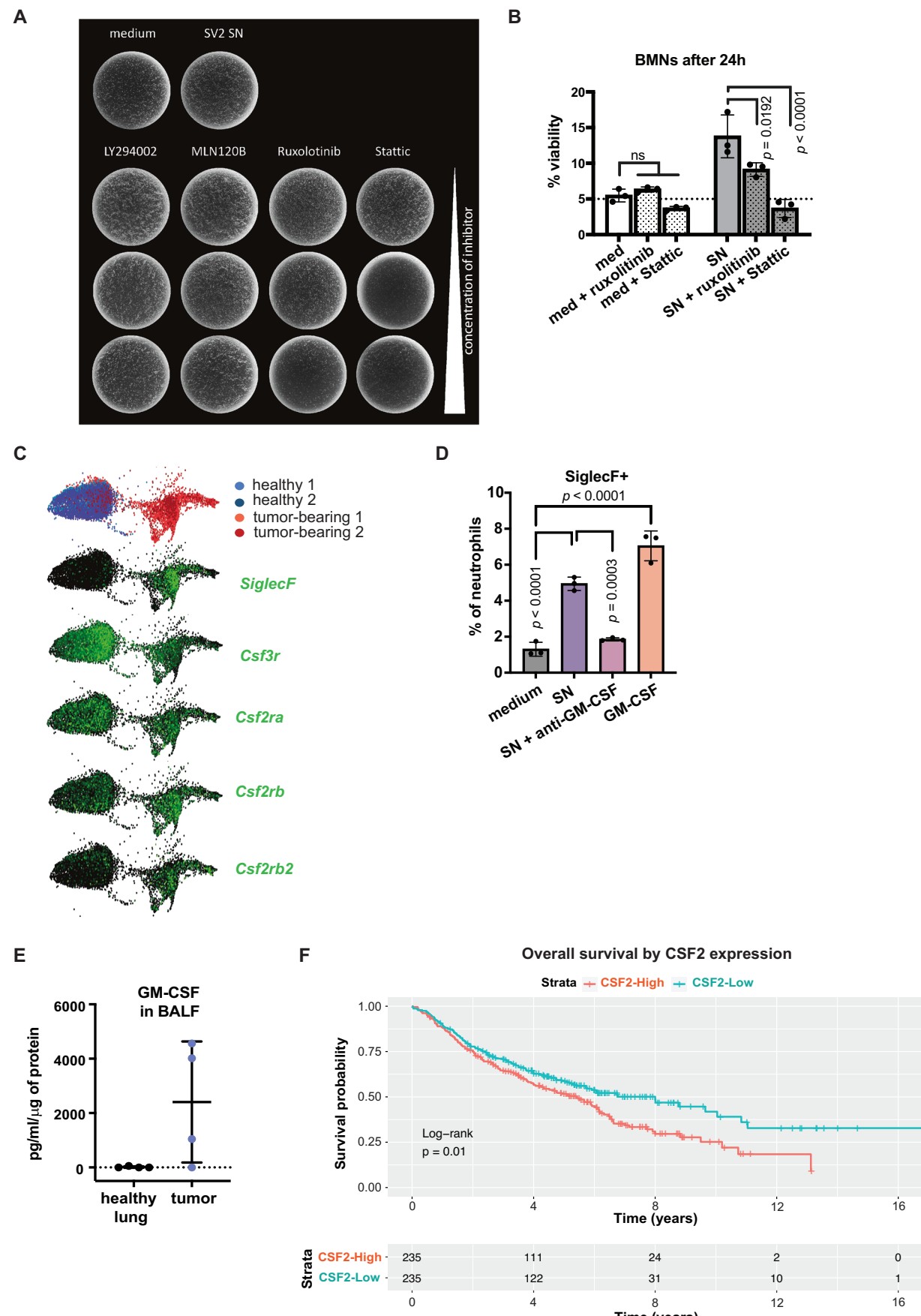

**A**

medium    SV2 SN

LY294002    MLN120B    Ruxolotinib    Stattic

concentration of inhibitor

**B**

BMNs after 24h

% viability

med
med + ruxolitinib
med + Stattic
SN
SN + ruxolitinib
SN + Stattic

ns

p = 0.0192

p < 0.0001

**C**

● healthy 1
● healthy 2
● tumor-bearing 1
● tumor-bearing 2

*SiglecF*

*Csf3r*

*Csf2ra*

*Csf2rb*

*Csf2rb2*

**D**

SiglecF+

p < 0.0001

% of neutrophils

medium
SN
SN + anti-GM-CSF
GM-CSF

p < 0.0001

p = 0.0003

**E**

GM-CSF
in BALF

pg/ml/µg of protein

healthy
lung
tumor

**F**

Overall survival by CSF2 expression

Strata  — CSF2-High  — CSF2-Low

Survival probability

Time (years)

Log−rank
p = 0.01

| Strata | | | | | |
|---|---|---|---|---|---|
| CSF2-High | 235 | 111 | 24 | 2 | 0 |
| CSF2-Low | 235 | 122 | 31 | 10 | 1 |
| | 0 | 4 | 8 | 12 | 16 |

Time (years)

◀  **Figure EV2.  Bcl-xL induction by GM-CSF-mediated JAK-STAT signaling.**

(**A**) Representative images of BMNs incubated with increasing doses of pathway inhibitors observed with brightfield microscopy. (**B**) BMN viability was measured by flow cytometry 24 h after incubation with indicated doses of ruxolitinib (1 µM) or stattic (10 µM) in medium or SV2 supernatant, with BMNs extracted from $n = 3$ mice. (**C**) Representative images of *SiglecF*, *Csf3r*, *Csf2ra*, *Csf2rb* and *Csf2rb2* from publicly available single-cell RNA sequencing data. (**D**) Percentage of SiglecF$^+$ BMNs after 24 h of incubation. $n = 3$ biological replicates. (**E**) GM-CSF concentration measured in the bronchoalveolar lavage fluid (BALF) from healthy ($n = 4$) and tumor-bearing mice ($n = 4$). (**F**) Kaplan–Meier curves for overall survival and *P* value of pairwise differences between groups with high or low *CSF2* expression from the combined LUAD transcriptome dataset. Data information: For (**B, D, E**), data are shown as mean ± SD. For (**B, D**), significance was determined by ordinary one-way ANOVA with Tukey's multiple comparisons test. ns non-significant.

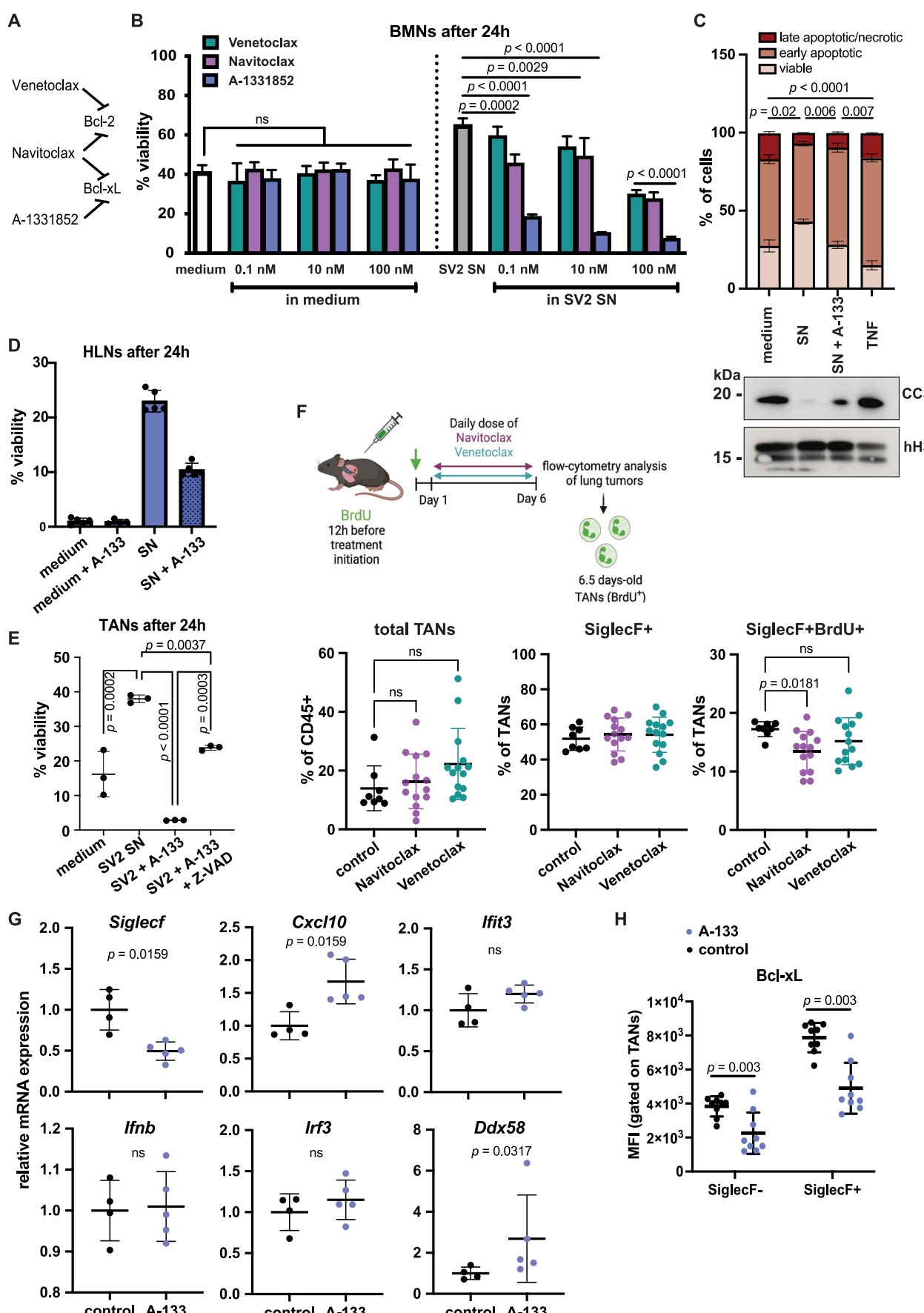

◀

**Figure EV3.  A-1331852 decreases TAN ageing.**

(A) Scheme representing BH3-mimetics inhibition specificity. (B) Viability (%) of BMNs incubated with 0.1, 10 and 100 nM of Venetoclax, Navitoclax or A-1331852 in medium or SV2 SN for 24 h. $n = 3$ biological replicates. (C) Upper part: percentage of viable (AnnexinV$^-$7-AAD$^-$), early (AnnexinV$^+$7-AAD$^-$), and late apoptotic (AnnexinV$^+$7-AAD$^+$) BMNs incubated with medium or SV2 SN with or without A-1331852. TNF (5 ng/mL) was used as control to induce neutrophil apoptosis. $n = 3$ biological replicates. Lower part: Western blot analysis of cleaved-caspase-3 (CC3). Histone H3 (hH3) was used as loading control. (D) Healthy lung neutrophils (HLN) survival after 24 h with SV2 SN and with A-1331852 (10 nM) (HLNs were extracted from $n = 5$ healthy non-tumor-bearing mice). (E) % of surviving TANs after 24 h, with SV2, SV2 + A-1331852 with or without preliminary incubation with the pan-caspase inhibitor z-VAD-FMK (20 μM). TANs are from $n = 3$ tumors. (F) Scheme showing the experimental design and plots showing flow cytometry analysis of neutrophils, SiglecF$^+$ and SiglecF$^+$BrdU$^+$ 6-days-old TANs in control KP mice ($n = 7$ tumors), mice treated with Navitoclax ($n = 12$ tumors) or with Venetoclax ($n = 14$). (G) Real-time PCR analysis of expression of the indicated genes in TANs extracted from $n = 4$ control or $n = 5$ A-1331852 treated tumors. (H) MFI of Bcl-xL expression in SiglecF$^-$ and SiglecF$^+$ TANs in mice from the same experiment as reported in Fig. 3E. $n = 9$ tumors for each group. Data information: All data are shown as mean ± SD. For (B), conditions with drugs in the medium were analyzed compared to the medium-only condition, and drugs in SV2 SN were compared to the SV2 SN condition and significance was determined by ordinary one-way ANOVA with Dunnett's multiple comparisons test. For (C), significance was based on two-way ANOVA with Tukey's multiple comparisons test. For (E), significance was determined by ordinary one-way ANOVA with Tukey's multiple comparisons test. For (F), total TANs were analyzed by Kruskal–Wallis with Dunn's multiple comparisons test. SiglecF$^+$BrdU$^+$ TANs were analyzed by ordinary one-way ANOVA and Dunn's multiple comparisons test. For (G), significance was based on a two-tailed Student's *t* test. For (H), significance was based on two-way ANOVA. ns non-significant.

**A**

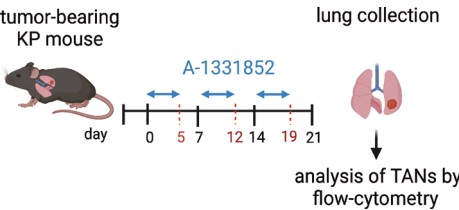

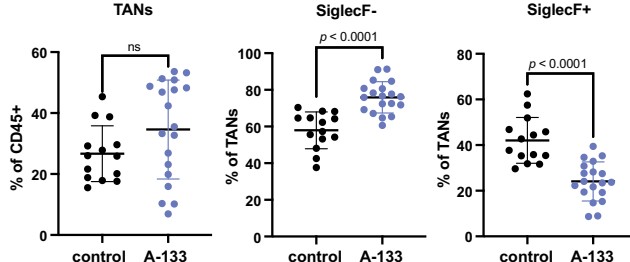

**Figure EV4. Intermittent A-1331852 treatment restores selective targeting against SiglecF⁺ TANs.**

(A) Scheme describing the treatment regimen and experimental setup. Mice were treated with A-1331852 for 5 days then with two days break, for a duration of 3 weeks. Lung tumors were then isolated and the TAN population was analyzed by flow cytometry. (B) Graphs showing the percentages of total TANs, SiglecF⁻ and SiglecF⁺ TANs. $n = 14$ tumors were analyzed for control mice (vehicle treated) and $n = 19$ tumors from A-133-treated mice. Data shown are mean ± SD. Significance was determined with a Mann–Whitney test for total TANs and two-tailed $t$ test for SiglecF⁻ and SiglecF⁺ percentages. ns non-significant.

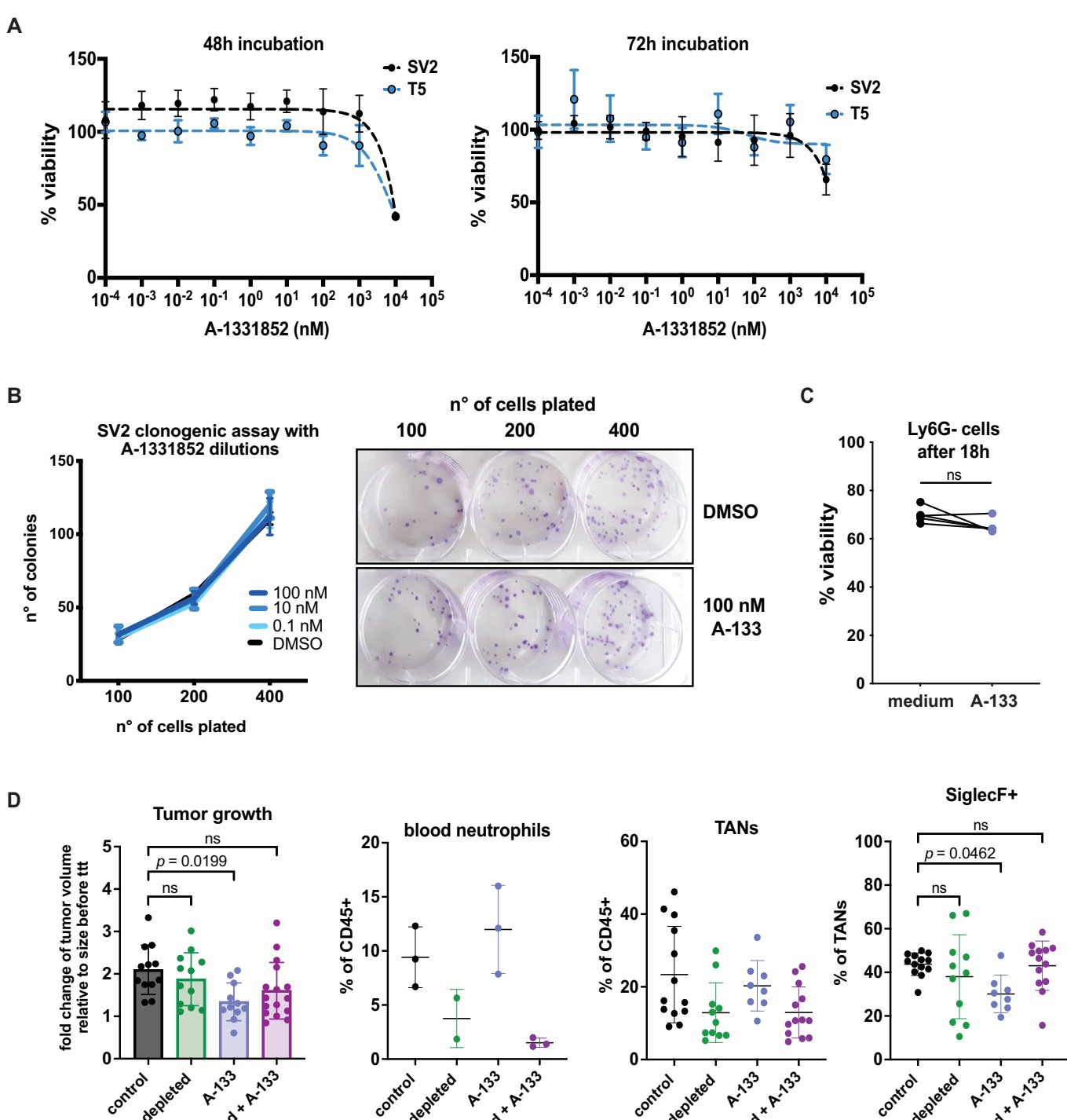

**Figure EV5. Bcl-xL blockade does not affect the viability of lung tumor cells.**

(A) Viability of SV2 and T5 cell lines, measured with PrestoBlue after 48 and 72 h of incubation with serial dilutions of A-1331852. $n = 3$ technical replicates. (B) Clonogenic assay performed with 100, 200, 400 single SV2 cells incubated with 0.1, 10 or 100 nM of A-1331852. $n = 3$ technical replicates. (C) Data show percentage of viable Ly6G- cells after 18 h of 10 nM of A-1331852 incubation ($n = 5$ tumors). (D) Plots showing the growth of single tumors in control ($n = 12$), A-1331852-treated ($n = 11$), neutrophil-depleted ($n = 12$) and neutrophil-depleted in combination with A-1331852 (d + A-133, $n = 16$) KP mice. TAN proportions out of total CD45+ and SiglecF+ cells out of total TANs are shown for single tumors for control ($n = 13$), A-1331852-treated ($n = 8$), neutrophil-depleted ($n = 10$) and neutrophil-depleted in combination with A-1331852 ($n = 13$). Blood neutrophil levels are also shown for $n = 3$ in control mice, A-1331852-treated and depletion with A-1331852, and $n = 2$ for neutrophil-depleted only mice. Data information: Data are shown as mean ± SD. For (C), significance was determined based on a paired $t$ test. For (D), ordinary one-way ANOVA with Tukey's multiple comparisons test was performed for the tumor growth and Kruskal–Wallis with Dunn's multiple comparisons test was performed for the other panels. ns, non-significant.

