## [Peer Review File · EMBO Molecular Medicine]

Bcl-xL targeting eliminates ageing tumor-promoting neutrophils and inhibits lung tumor growth

Anita Bodac, Abdullah Mayet, Sarika Rana, Justine Pascual, Amber Bowler, Vincent Roh, Nadine Fournier, Ligia Craciun, Pieter Demetter, Freddy Radtke, and Etienne Meylan

DOI: [10.15252/emmm.202318237](https://doi.org/10.15252/emmm.202318237)

Corresponding author(s): Etienne Meylan (etienne.meylan@ulb.be)

Review Timeline:

Submission Date:	28th Jun 23
Editorial Decision:	19th Jul 23
Revision Received:	27th Oct 23
Editorial Decision:	14th Nov 23
Revision Received:	24th Nov 23
Accepted:	28th Nov 23

Editor: Lise Roth

Transaction Report:

19th Jul 2023

Dear Prof. Meylan,

Thank you for the submission of your manuscript to EMBO Molecular Medicine. We have now received feedback from the three reviewers who agreed to evaluate your manuscript. As you will see from the reports below, the referees acknowledge the interest of the study and are overall supporting publication of your work pending appropriate revisions.

Addressing the reviewers' concerns in full will be necessary for further considering the manuscript in our journal, and acceptance of the manuscript will entail a second round of review. Following further discussion with the referees, we would like to clarify that additional studies with the combo therapy (Bcl-xL inhibitor + ICB) will NOT be required for further consideration of your manuscript, but we would rather ask you to include a discussion on why the combination treatment did not work.

EMBO Molecular Medicine encourages a single round of revision only and therefore, acceptance or rejection of the manuscript will depend on the completeness of your responses included in the next, final version of the manuscript. For this reason, and to save you from any frustrations in the end, I would strongly advise against returning an incomplete revision.

Revised manuscripts should be submitted within three months of a request for revision; they will otherwise be treated as new submissions, except under exceptional circumstances in which a short extension is obtained from the editor.

We require:

4) A .docx formatted letter INCLUDING the reviewers' reports and your detailed point-by-point responses to their comments. As part of the EMBO Press transparent editorial process, the point-by-point response is part of the Review Process File (RPF), which will be published alongside your paper.

5) A complete author checklist, which you can download from our author guidelines (<https://www.embopress.org/page/journal/17574684/authorguide#submissionofrevisions>). Please insert information in the checklist that is also reflected in the manuscript. The completed author checklist will also be part of the RPF.

6) It is mandatory to include a 'Data Availability' section after the Materials and Methods. Before submitting your revision, primary datasets produced in this study need to be deposited in an appropriate public database, and the accession numbers and database listed under 'Data Availability'. Please remember to provide a reviewer password if the datasets are not yet public (see <https://www.embopress.org/page/journal/17574684/authorguide#dataavailability>).

In case you have no data that requires deposition in a public database, please state so in this section ("This study includes no data deposited in external repositories."). Note that the Data Availability Section is restricted to new primary data that are part of this study.

7) For data quantification: please specify the name of the statistical test used to generate error bars and P values, the number (n) of independent experiments (specify technical or biological replicates) underlying each data point and the test used to calculate p-values in each figure legend. The figure legends should contain a basic description of n, P and the test applied. Graphs must include a description of the bars and the error bars (s.d., s.e.m.). Please provide exact p values.

8) Our journal encourages inclusion of *data citations in the reference list* to directly cite datasets that were re-used and obtained from public databases. Data citations in the article text are distinct from normal bibliographical citations and should

directly link to the database records from which the data can be accessed. In the main text, data citations are formatted as follows: "Data ref: Smith et al, 2001" or "Data ref: NCBI Sequence Read Archive PRJNA342805, 2017". In the Reference list, data citations must be labeled with "[DATASET]". A data reference must provide the database name, accession number/identifiers and a resolvable link to the landing page from which the data can be accessed at the end of the reference. Further instructions are available at .

9) We replaced Supplementary Information with Expanded View (EV) Figures and Tables that are collapsible/expandable online. A maximum of 5 EV Figures can be typeset. EV Figures should be cited as 'Figure EV1, Figure EV2" etc... in the text and their respective legends should be included in the main text after the legends of regular figures.

10) The paper explained: EMBO Molecular Medicine articles are accompanied by a summary of the articles to emphasize the major findings in the paper and their medical implications for the non-specialist reader. Please provide a draft summary of your article highlighting

11) For more information: There is space at the end of each article to list relevant web links for further consultation by our readers. Could you identify some relevant ones and provide such information as well? Some examples are patient associations, relevant databases, OMIM/proteins/genes links, author's websites, etc...

12) Author contributions: CRediT has replaced the traditional author contributions section because it offers a systematic machine readable author contributions format that allows for more effective research assessment. Please remove the Authors Contributions from the manuscript and use the free text boxes beneath each contributing author's name in our system to add specific details on the author's contribution. More information is available in our guide to authors.

13) Disclosure statement and competing interests: We updated our journal's competing interests policy in January 2022 and request authors to consider both actual and perceived competing interests. Please review the policy <https://www.embopress.org/competing-interests> and update your competing interests if necessary.

14) Every published paper now includes a 'Synopsis' to further enhance discoverability. Synopses are displayed on the journal webpage and are freely accessible to all readers. They include a short stand first (maximum of 300 characters, including space) as well as 2-5 one-sentences bullet points that summarizes the paper. Please write the bullet points to summarize the key NEW findings. They should be designed to be complementary to the abstract - i.e. not repeat the same text. We encourage inclusion of key acronyms and quantitative information (maximum of 30 words / bullet point). Please use the passive voice. Please attach these in a separate file or send them by email, we will incorporate them accordingly.

15) As part of the EMBO Publications transparent editorial process initiative (see our Editorial at <http://embomolmed.embopress.org/content/2/9/329>), EMBO Molecular Medicine will publish online a Review Process File (RPF) to accompany accepted manuscripts.

In the event of acceptance, this file will be published in conjunction with your paper and will include the anonymous referee reports, your point-by-point response and all pertinent correspondence relating to the manuscript. Let us know whether you agree with the publication of the RPF and as here, if you want to remove or not any figures from it prior to publication.

I look forward to receiving your revised manuscript.

Yours sincerely,

Lise Roth

***** Reviewer's comments *****

Referee #1 (Remarks for Author):

This study by Bodac et al. investigates a novel therapeutic approach to target tumor-associated Neutrophils (TANs) that are associated with poor prognosis and immune evasion. The authors identify that GM-CSF promotes the expression of Bcl-xL that in turn enhances TAN survival through JAK/STAT signaling. In addition, they observe that SiglecF positive TANs, which are found in lung tumors, have increased levels of Bcl-xL. They demonstrate that targeting Bcl-xL therapeutically suppresses TAN survival. This is a very interesting and timely study that provides important insight into the role of young and old neutrophils during tumorigenesis. This study will open new avenues of investigation in better understanding the complex nature of neutrophils in both promoting tumor growth and suppressing anti-tumor immune responses.

I have a few points that may need to be addressed in order to improve this manuscript:

- 1) The in vitro culture of BMNs or TANs with tumor cell supernatant offers an opportunity to also investigate the relationship between GM-CSF, STAT signaling, Bcl-xL and SiglecF. Do the authors see any difference in cell surface SiglecF in the different conditions in Fig1 and 2? Does GM-CSF and STAT-signaling play any role in the regulation of SiglecF expression?
- 2) Given that Bcl-xL is an anti-apoptotic protein, the authors should demonstrate that inhibition of Bcl-xL with A-133 leads to apoptosis by staining for Cleaved-caspase 3 by IF and/or FACS. Their invitro studies with the apoptosis suppressor z-VAD in Figure S3D show a minimal rescue in viability in the presence of A-133. How do the authors explain this?
- 3) Dissecting the tumor intrinsic effects of Bcl-xL inhibition would be very interesting. Do KP tumors express Bcl-xL and what is the effect of treating KP cells with the Bcl-xL inhibitor A-133? The authors provide ki67 staining data from the tumors, however, it is unclear whether this is coming from the tumor cells or immune cells in the microenvironment. Immunofluorescence or IHC co-staining with a tumor cell marker would be helpful. In addition, the authors should check apoptosis markers such as cleaved caspase-3 by IHC.

Referee #2 (Comments on Novelty/Model System for Author):

The model is adequate.

To improve technical quality:

- Information on Bcl-2 family protein expression other than Bcl-xL should be included.
- Validation of MPO as a marker of tissue neutrophils (as opposed to MPO expression by a subset of tissue macrophages).
- Information on patient characteristics should be included.
- In some experiments, the n values (2 or 3) are low, though the results are critical to support the translational potential of Bcl-xL inhibition.

This manuscript reports on the exciting possibility of selective targeting of tumor-associated neutrophils through inhibition of Bcl-xL to control growth of lung carcinoma in a mouse model. Overall, the experiments are well-designed, properly controlled and powered, and the experimental data lend support to the main conclusions. The manuscript is of interest, but the authors need to address some specific points to increase the quality of the work.

Referee #2 (Remarks for Author):

Bodac and colleagues report that pharmacological targeting of Bcl-xL activity decreased the survival and abundance of tumor-associated predominantly SiglecF⁺ neutrophils (TANs) and reduced tumor growth without causing neutropenia in a genetically engineered mouse model of lung adenocarcinoma. They show that tumor cell-derived GM-CSF induces the expression of Bcl-xL protein and prolongs neutrophil survival through the JAK/STAT signaling pathway. The study combines human transcriptomics, mouse bulk and human single-cell mRNAseq analyses, in vitro experiments, experiments in a mouse lung cancer model and immunohistochemical analysis of archived human lung cancer specimens. The authors also report that treatment with G-CSF increased the proportion of young TANs and augmented the anti-tumor activity of Bcl-xL blockade. The findings lend support to the intriguing possibility of selective targeting the aberrant longevity (and likely the activity) of TANs for the treatment of solid cancers, and are of general interest. I have the following comments for the authors' consideration to increase the quality of their work.

Major Comments.

1. Previous studies showed the anti-tumor/tumor-promoting activity of neutrophils may depend on the size of the tumor, indicating reprogramming TANs within the cancer microenvironment. SiglecF⁺ TANs expressed higher levels Bcl-xL levels than their SiglecF⁻ counterparts. However, A-1331852 produced marked reductions in the number of both SiglecF⁺ and SiglecF⁻ TANs (Fig.4E). How could the actions of SiglecF⁻ TANs be reconciled with low Bcl-xL expression? Would elimination of SiglecF⁻ TANs be associated with increased cancer growth?
2. While MPO is widely used as a neutrophil marker, a subset of tissue macrophages also expresses this protein. Thus, additional markers will be needed to validate the co-localization studies by excluding the possibility of upregulation of Bcl-xL in macrophages in KP tumors.
3. Three out of four tumor-bearing mice exhibited elevated GM-CSF levels. Were any differences in Bcl-xL expression observed in these mice?
4. What was the rationale for using various durations for treatment with A-1331852? In particular, why was 8-day duration chosen for the single dose BrdU experiments to address neutrophil survival?
5. The concept of A-1331852 sensitization to immune checkpoint inhibition is interesting and clinically relevant. However, the results at the very best may indicate trends, hence not convincing. Indeed, the main text and title of Suppl. Fig. 5 appears to be discordant. Since this line of studies adds very little, if anything, to the central concept of the manuscript, this reviewer would suggest removing the data and discussion from the current manuscript.
6. Based on bulk and single cell RNAseq, the authors focus on Bcl-xL. What was the expression level of other Bcl-2 family proteins, including Mcl-1 or A2 (which are known to regulate neutrophil life span)? Arguably, these proteins could also contribute to aberrant survival of TANs, and hence may limit the long-term use of anti-Bcl-xL therapy.
7. Some information should be provided on patient characteristics (e.g. type of cancer, duration of disease, treatment, sex and age) for the archived specimens.
8. Suppl. Fig. 4C. The number of mice (n=2 or 3) for analysis of blood neutrophils is too low to make meaningful comparisons. The circulating number of neutrophils is considerably lower in mice than humans, and neutrophils are rapidly mobilized from the bone marrow in response to inflammatory stimuli in mice. Could the author comment on the effect of A-1331852 on neutrophil mobilization?

Minor Points

1. Lines 298 and 303. Please insert "Ab" after "anti-rat".
2. Many references are incomplete. Volume and page numbers should be presented.
3. Fig. 2A and B lacks statistical comparisons.
4. Fig. 6E. Viability is expressed relative to medium only. Thus, a 3-fold increase is difficult to interpret in A549 SN only. Expressing viability in percentage should be more informative.

Referee #3 (Comments on Novelty/Model System for Author):

The technical quality of the experiments displayed is adequate. The authors perform the necessary controls and the endpoints chosen are reasonable. I think the novelty of this manuscript is high. Neutrophils are known to be plastic in the context of cancer and it is appreciated that some tumor associated neutrophils promote tumor progression. However, details regarding which neutrophil subsets are tumor-promoting are still uncertain and the biology of these neutrophils is an active area of research. This study provides new insight into TANs and TAN biology. Further it provides a feasible/attractive strategy to selectively target tumor-promoting TANs. Thus, the potential impact is also high. The studies exploit a GEMM of NSCLC and human tissue, these are adequate models to address the major queries of the study.

Referee #3 (Remarks for Author):

Bodac et al present a study that nominates targeting Bcl-xL high tumor associated neutrophils (TANs) as an intriguing novel therapeutic strategy for lung cancer. The introduction section provides an effective framework for the study. TAN biology is an

area of active investigation by many groups. Queries of interest include: the heterogeneity of neutrophil phenotypes in tumors, whether distinct subsets of TANs can be associated with tumor promotion and how to effectively and safely target tumor promoting TANs. Bodac and colleagues provide potential insight into these questions. The authors note that some neutrophils survive longer in tumors than anticipated and that there are conditions/pathologies where neutrophil apoptosis is delayed. These observations induce the authors to investigate the expression of genes known to modulate survival or apoptosis. This results in identification of Bcl2l1 (Bcl-xl) as a gene/protein with elevated expression in TANs compared to neutrophils in healthy lung (HLN). The authors subsequently provide evidence that Bcl-xl expression can be induced by GM-CSF produced by tumor cells and that GM-CSF induction of Bcl-xl is dependent upon JAK/STAT activity. They also provide evidence that Bcl-xl expression correlates with neutrophil expression of SiglecF, a marker that has been associated with tumor-promoting neutrophils. The authors present the surprising finding that selective inhibition of Bcl-xl with A-1331852 reduces the viability of TANs and SiglecF+ neutrophils but not bone marrow-derived neutrophils. The authors find that treatment of mice bearing KP lung tumors with A-1331852 reduces tumor growth and reduces the number of aged neutrophils in the tumor microenvironment. Thus nominating targeting aged TANs by inhibiting Bcl-xl as a novel strategy for lung cancer.

The manuscript is presented clearly and the data displayed in general support the conclusions drawn. The comments below are largely minor in nature.

Comments:

1. HLN express higher levels of Bcl-xl than neutrophils in the bone marrow or blood. Do HLN survive longer than neutrophils in other organs/systems?
2. The flow data displayed in Figure 3&4 is convincing that treatment of A-1331852 alters the immune landscape. Yet the authors note that combining A-1331852 with anti-PD-1 was not an effective combination for controlling tumor progression (SFig5). Is there an observable change in T cells (Cd8 & Cd4) and Tregs in terms of location in the tumor after treatment with A-1331852, I wonder if multi-plex IHC on tumor sections might reveal that while T cell numbers were increased, effector T cells are still excluded from tumor cell nests, thus potentially explaining in part the lack of effective combo therapy?

Referee #1 (Remarks for Author):

This study by Bodac et al. investigates a novel therapeutic approach to target tumor-associated Neutrophils (TANs) that are associated with poor prognosis and immune evasion. The authors identify that GM-CSF promotes the expression of Bcl-xL that in turn enhances TAN survival through JAK/STAT signaling. In addition, they observe that SiglecF positive TANs, which are found in lung tumors, have increased levels of Bcl-xL. They demonstrate that targeting Bcl-xL therapeutically suppresses TAN survival. This is a very interesting and timely study that provides important insight into the role of young and old neutrophils during tumorigenesis. This study will open new avenues of investigation in better understanding the complex nature of neutrophils in both promoting tumor growth and suppressing anti-tumor immune responses.

I have a few points that may need to be addressed in order to improve this manuscript:

1) The *in vitro* culture of BMNs or TANs with tumor cell supernatant offers an opportunity to also investigate the relationship between GM-CSF, STAT signaling, Bcl-xL and SiglecF. Do the authors see any difference in cell surface SiglecF in the different conditions in Fig1 and 2? Does GM-CSF and STAT-signaling play any role in the regulation of SiglecF expression?

We thank the Reviewer for all the comments and suggestions.

We agree about the opportunity our *in vitro* culture system offers to investigate relationships between GM-CSF, STAT signaling and SiglecF in neutrophils. Using the same experimental setting, we have thus performed additional experiments, with each of the tumor cell SN and GM-CSF to induce SiglecF cell surface expression, adding (or not) different concentrations of the STAT inhibitor, Stattic. Flow cytometry analyses demonstrate that SiglecF upregulation in response to tumor SN or GM-CSF is STAT-dependent. These data, which complement our previous Figure 2E by adding information about STAT inhibition, are now included in main text p.9 and as **new Figure 2E**. Instead, our former Figure 2E has been placed in the Expanded View (Figure EV2D).

Next, to measure if SiglecF induction may occur transcriptionally, we performed real-time PCR analyses from neutrophils cultured in tumor SN or GM-CSF ± Stattic, and analysed *SiglecF* gene expression. These data show induction of *SiglecF* in response to each of SN and GM-CSF, and its repression upon STAT inhibition in both cases. However, the responses were heterogeneous, and the results obtained with Stattic were just not significant. Thus, we present these data, suggesting a transcriptional control of SiglecF, for Reviewer here below:

***SiglecF* induction by tumor cell SN and GM-CSF is repressed by STAT inhibition.** Graph representing fold change in *SiglecF* mRNA expression in BMNs 24 h post treatment with SV2 conditioned media or GM-CSF (10 ng/ml) in presence of vehicle (DMSO) or Stattic (1 μM) with respect to BMNs in control medium. Data shown are means ± S.D. from four independent experiments.

2) Given that Bcl-xL is an anti-apoptotic protein, the authors should demonstrate that inhibition of Bcl-xL with A-133 leads to apoptosis by staining for Cleaved-caspase 3 by IF and/or FACS. Their *in vitro* studies with the apoptosis suppressor z-VAD in Figure S3D show a minimal rescue in viability in the presence of A-133. How do the authors explain this?

In response to the first part of this comment, we have now specifically monitored apoptosis. First, using Annexin-V + 7-AAD staining, we demonstrate that the tumor SN decreases the percentage

of apoptotic neutrophils compared to medium, whereas A-133 enhances apoptosis in SN culture conditions. TNF was used as a positive control for neutrophil apoptosis. Second, we monitored the cleavage of caspase-3 (CC3) by Western blot in the same culture conditions. This shows that tumor SN attenuates neutrophil apoptosis, while A-133 increases apoptosis in tumor SN conditions. Again, TNF induced the cleavage of caspase-3. The results from the flow cytometry and Western blot have been placed together, one above the other, and constitute a new Figure panel: **new Figure EV3C**. The text has been updated on p.11.

Concerning the puzzling, minimal rescue obtained with Z-VAD in Fig. S3D, we first reasoned that its concentration was not optimal and repeated the experiment with different doses. However, this did not improve the rescue (not shown). Next, instead of adding Z-VAD and A-133 together, we decided to incubate neutrophils first with Z-VAD followed, 1 hour later, by A-133 treatment. By adding this pre-incubation time, we monitor a better rescue of A-133-mediated cell death by Z-VAD. Because this constitutes an optimized protocol, we have replaced the former Figure panel by a new Figure panel, now becoming **Figure EV3E**.

3) Dissecting the tumor intrinsic effects of Bcl-xL inhibition would be very interesting. Do KP tumors express Bcl-xL and what is the effect of treating KP cells with the Bcl-xL inhibitor A-133? The authors provide ki67 staining data from the tumors, however, it is unclear whether this is coming from the tumor cells or immune cells in the microenvironment. Immunofluorescence or IHC co-staining with a tumor cell marker would be helpful. In addition, the authors should check apoptosis markers such as cleaved caspase-3 by IHC.

We agree with Reviewer that blocking Bcl-xL directly in the tumor cells could be very interesting, too, as it could contribute to the antitumor effect of A-133. By IHC, we find a variable expression of Bcl-xL in KP tumors, with some tumors showing no detectable protein and others no-to-weak expression. An example is shown below for the Reviewer:

Our data, however, do not currently support that A-133 affects the viability of KP tumor cells when used as single agent. Indeed, using two KP cell lines *in vitro* we failed to detect any cell death or reduction in colony formation by A-133 even at high doses (Fig. S4A, S4B, now Fig. EV5A and EV5B). This agrees with REF Potter *et al.*, where the authors showed that Bcl-xL blockade, although not killing NSCLC cell lines when used alone, increased their mitochondrial priming, sensitizing them to chemotherapy. Thus, for a future study we plan to use A-133 in combination with different chemotherapies used in lung cancer treatment, to evaluate if additive or synergistic anti-tumor effects could be obtained. We believe, however, that such combinatorial therapy investigations go beyond the scope of the present manuscript.

Nevertheless, to monitor if other cells from the tumor microenvironment – including tumor cells – are sensitive to Bcl-xL blockade, we isolated KP tumors, prepared single cell suspensions and removed TANs with an anti-Ly6G antibody coupled to magnetic beads. We subjected the Ly6G negative tumor-derived cell fraction (composed mostly of tumor cells, but also other immune cells and stromal cells) to A-133 (10 nM) and monitored viability 18h later. We monitored no significant reduction in Ly6G- cell survival. This data supports the notion that the main targets of A-133 are TANs and, more selectively, ageing TANs. We add this new data as **new Figure EV5C**. The text has been updated on p.14.

To clarify if Ki67 staining in tumors came from tumor cells or immune cells, we have performed a flow cytometry analysis of the tumor-associated immune cells. This reveals some immune cell proliferation, as immune (CD45+) cells positive for Ki67 accounted for ~20-30% of all tumor-associated immune cells, as assessed using flow cytometry. In the presence of A-1331852, there was no diminished immune cell proliferation. We therefore deduct that the decrease in Ki67+ measured by IHC reflects the decrease in proliferation of tumor cells.

Finally, we have performed a double immunohistochemistry with an anti-cleaved caspase-3 (CC3) antibody together with anti-pan-cytokeratin (staining tumor epithelial cell). However, we almost never detect any tumor cell that is positive for CC3, in control or in A-133 treated conditions (typically 2-5 CC3 positive cells per tumor in each condition). Thus, tumor cell apoptosis – at least inferred from CC3 staining – does not seem to be a major response to the treatment. We show below, for the Reviewer, a few examples of the co-staining obtained. CC3 is indicated with arrowheads in the magnified images.

Referee #2 (Comments on Novelty/Model System for Author):

The model is adequate.

To improve technical quality:

- Information on Bcl-2 family protein expression other than Bcl-xL should be included.
- Validation of MPO as a marker of tissue neutrophils (as opposed to MPO expression by a subset of tissue macrophages).
- Information on patient characteristics should be included.
- In some experiments, the n values (2 or 3) are low, though the results are critical to support the translational potential of Bcl-xL inhibition.
- Information on Bcl-2 family protein expression other than Bcl-xL should be included.

This manuscript reports on the exciting possibility of selective targeting of tumor-associated neutrophils through inhibition of Bcl-xL to control growth of lung carcinoma in a mouse model.

Overall, the experiments are well-designed, properly controlled and powered, and the experimental data lend support to the main conclusions. The manuscript is of interest, but the authors need to address some specific points to increase the quality of the work.

We thank the Reviewer for all the comments and suggestions.
The above points are addressed below in the "Remarks for Author".

Referee #2 (Remarks for Author):

Bodac and colleagues report that pharmacological targeting of Bcl-xL activity decreased the survival and abundance of tumor-associated predominantly SiglecF⁺ neutrophils (TANs) and reduced tumor growth without causing neutropenia in a genetically engineered mouse model of lung adenocarcinoma. They show that tumor cell-derived GM-CSF induces the expression of Bcl-xL protein and prolongs neutrophil survival through the JAK/STAT signaling pathway. The study combines human transcriptomics, mouse bulk and human single-cell mRNAseq analyses, in vitro experiments, experiments in a mouse lung cancer model and immunohistochemical analysis of archived human lung cancer specimens. The authors also report that treatment with G-CSF increased the proportion of young TANs and augmented the anti-tumor activity of Bcl-xL blockade. The findings lend support to the intriguing possibility of selective targeting the aberrant longevity (and likely the activity) of TANs for the treatment of solid cancers, and are of general interest. I have the following comments for the authors' consideration to increase the quality of their work.

Major Comments.

1. Previous studies showed the anti-tumor/tumor-promoting activity of neutrophils may depend on the size of the tumor, indicating reprogramming TANs within the cancer microenvironment. SiglecF⁺ TANs expressed higher levels Bcl-xL levels than their SiglecF⁻ counterparts. However, A-1331852 produced marked reductions in the number of both SiglecF⁺ and SiglecF⁻ TANs (Fig.4E). How could the actions of SiglecF⁻ TANs be reconciled with low Bcl-xL expression? Would elimination of SiglecF⁻ TANs be associated with increased cancer growth?

We thank the Reviewer for the insightful remark, which we took to heart. Targeting some SiglecF⁻ TANs was a possible risk, as these cells are positive for Bcl-xL, even if it is expressed at lower levels than in SiglecF⁺ TANs (see Figure 1D). Our initial goal for treating KP mice with increasing time durations of A-133 (1, 2 and 3 weeks) was precisely to assess the targeting selectivity toward SiglecF⁺ TANs over time. Specifically, we wanted to know if SiglecF⁺ TAN targeting would diminish over time (=decreased efficacy) or if more TANs would become targeted (=decreased selectivity; targeting of SiglecF⁺ but also SiglecF⁻ cells, resulting in diminished total TANs). After three weeks of treatment, we observed the latter, as noticed by the Reviewer. Thus, we agree with Reviewer that long-term treatment, which not only eliminates SiglecF⁺ but also decreases the abundance of SiglecF⁻ and total TANs, could limit treatment efficacy.

To address this loss of selectivity issue, we used the following approach: based on our initial data obtained with long-term treatment using daily A-133 injections, we reasoned that the introduction of a treatment pausing would relieve the constant pressure imposed on ageing TANs, enabling them to increase their lifespan a little, before treatment re-start. Our hope was that such a treatment (which we decided to set as 5 days ON, 2 days OFF, for a total of 3 weeks) would restore the selective targeting of SiglecF⁺ TANs without impacting total TANs. Our new, intermittent treatment protocol was used in KP mice, after which we isolated the tumor-associated immune cells (CD45⁺). In contrast to our previous data obtained after long-term uninterrupted treatment, the abundance of total TANs was not changed upon intermittent A-133 treatment compared to control tumors. However, the SiglecF⁻ and the SiglecF⁺ TAN subsets increased and decreased very significantly, respectively, in response to intermittent A-133. Thus, this new treatment protocol enabled to restore the selective targeting of old, SiglecF⁺ TANs while preserving younger cells. We have added this new data as **new Figure EV4A-B**. The text has been updated on p.14.

2. While MPO is widely used as a neutrophil marker, a subset of tissue macrophages also expresses this protein. Thus, additional markers will be needed to validate the co-localization studies by excluding the possibility of upregulation of Bcl-xL in macrophages in KP tumors.

We agree with Reviewer that some tissue macrophages might express MPO. However, interrogating *Bcl2l1* gene expression in all immune cells from KP tumors (REF Zilionis *et al.*) reveals that it is almost absent from macrophages. A screenshot of this is shown below for the Reviewer.

Nevertheless, to validate our findings we have performed a co-IF using anti-MPO with anti-Ly6G, which specifically stains neutrophils. Our data reveal a strong co-localization between MPO and Ly6G expression in KP tumors. We show some examples of the co-IF images and magnifications below for the Reviewer.

Together, these data strongly support that Bcl-xL staining in tumors comes mainly from TANs.

Representative images of immunofluorescence staining of TANs identified with MPO (green) and Ly6G (red) in KP tumors. The scale bar is 20 µm.

3. Three out of four tumor-bearing mice exhibited elevated GM-CSF levels. Were any differences in Bcl-xL expression observed in these mice?

In this experiment, we had isolated and pooled TANs from several tumors per mouse and had extracted RNA for reverse transcription. In response to the Reviewer's question, we have thus performed a real-time PCR analysis from these pooled TAN cDNAs, which shows a positive correlation between GM-CSF in BALF from the four tumor-bearing mice and Bcl-xL expression in TANs. We provide the data for the Reviewer here below:

GM-CSF in BALF correlates with Bcl2l1 expression in TANs.
Pearson correlation analysis of GM-CSF levels in the bronchoalveolar lavage fluid (BALF) of tumor-bearing mice and Bcl2l1 expression levels (relative to reference gene) in TANs isolated from multiple tumors from the corresponding mouse. Data are shown for $n = 4$ tumor-bearing mice.

4. What was the rationale for using various durations for treatment with A-1331852? In particular, why was 8-day duration chosen for the single dose BrdU experiments to address neutrophil survival?

The rationale was to monitor the evolution of the response to A-133 treatment. In particular, after we had performed the short (8-day) treatment, which successfully showed a decreased SiglecF+ TANs and concomitant increased SiglecF- TANs, we wanted to know if such selectivity against old TANs would be maintained or lost over time. See also our response to your major comment #1 above, resulting in the establishment of a new scheme achieving long-term selective SiglecF+ TAN targeting.

There was not a strong reason for choosing to use BrdU only for the shortest, 8-day treatment. Because we wanted our readout at the shortest treatment duration to be flow cytometry to monitor changes in TANs, we decided to use BrdU. At longer time points, we thought we would monitor other tumor parameters where BrdU staining would not be necessarily needed. At the end we were also able to collect tumors for flow cytometry at 2- and 3-week treatment.

5. The concept of A-1331852 sensitization to immune checkpoint inhibition is interesting and clinically relevant. However, the results at the very best may indicate trends, hence not convincing. Indeed, the main text and title of Suppl. Fig. 5 appears to be discordant. Since this line of studies adds very little, if anything, to the central concept of the manuscript, this reviewer would suggest removing the data and discussion from the current manuscript.

Our results show that there is no sensitization to anti-PD1 by A-1331852 (Fig. S5, now **Appendix Figure S2**). Thus, we believe, respectfully, that they fit with our main text "Combination treatment revealed ... no improvement compared to A-1331852 alone" and with the title of the legend "Bcl-xL blockade does not sensitize KP tumors to anti-PD1".

Because the Editor asked us to discuss why the combination treatment did not work, we have not removed these data, but have placed them instead in the Appendix, as **Appendix Figure S2**.

6. Based on bulk and single cell RNAseq, the authors focus on Bcl-xL. What was the expression level of other Bcl-2 family proteins, including Mcl-1 or A2 (which are known to regulate neutrophil life span)? Arguably, these proteins could also contribute to aberrant survival of TANs, and hence may limit the long-term use of anti-Bcl-xL therapy.

To interrogate the expression of other Bcl-2 family members in TANs, we have now analyzed transcript levels from the following anti-apoptotic Bcl-2 family genes: *Bcl2*, *Mcl1*, *Bfl-1 (Bcl2a1)*, in

our bulk RNAseq and in the different neutrophil subsets identified in REF Zilionis *et al.* from both human and mouse:

- From our bulk RNAseq, we did not detect *Bcl2*, as its expression is probably too low in HLN and in TANs; *Mcl1* was more expressed in HLN compared to TANs; *Bcl2a1b* was, like *Bcl2l1* but with a lower p-value, more expressed in TANs compared to HLN. We have now replaced our former Volcano plot (Fig. S1B), which only highlighted *Bcl2l1*, by a new one where *Bcl2l1*, *Mcl1* and *Bcl2a1b* are all indicated (**new Figure EV1B**). The text has been updated on p.6-7.

- From analyzing the mouse data of REF Zilionis *et al.*, we provide the expression of *Bcl2* (again almost absent from neutrophils), *Mcl1* and *Bcl2a1b* in addition to *Bcl2l1*. This shows high expression of *Mcl1* across neutrophil subsets, with higher expression in N1, and an expression of *Bcl2a1b* that resembles that of *Bcl2l1* and SiglecF. We have replaced the former Fig. S1D by **new Figure EV1E**. The text has been updated on p.7.

- From analyzing the human data of REF Zilionis *et al.*, we provide the expression of *BCL2* (weakly expressed in neutrophils, similarly to the mouse), *MCL1* and *BCL2A1* in addition to *BCL2L1*. This shows high expression of *MCL1* in most neutrophil subsets, with higher expression in N1, and high expression of *BCL2A1* in most neutrophil subsets, with higher expression in N5. We have replaced the former Fig. 6A by **new Figure 6A**. The text has been updated on p.16.

- Finally, we have performed a real-time PCR analysis of *Bcl2*, *Bcl2l1*, *Bcl2a1* and *Mcl1*, to compare the expression of each gene between HLN (n= 5 samples) and TANs (n= 11 samples). Specifically, *Bcl2l1* expression showed a significant difference between HLN and TANs, with an increased expression in the latter. Agreeing with the Volcano plot from Figure EV1B, the expression of *Bcl2a1* and *Mcl1* was higher and lower, respectively, in TANs compared to HLN, however these were only trends. This data is now included as **new Figure EV1C**. The text has been updated on p.6-7.

7. Some information should be provided on patient characteristics (e.g. type of cancer, duration of disease, treatment, sex and age) for the archived specimens.

We have now provided a Table containing all known clinical information about the patients for the archived specimens. This is provided in the Appendix PDF file and called "Appendix Table S1". In the main text, this Table is mentioned on page 16.

8. Suppl. Fig. 4C. The number of mice (n=2 or 3) for analysis of blood neutrophils is too low to make meaningful comparisons. The circulating number of neutrophils is considerably lower in mice than humans, and neutrophils are rapidly mobilized from the bone marrow in response to inflammatory stimuli in mice. Could the author comment on the effect of A-1331852 on neutrophil mobilization?

We agree that the number of tumor-bearing mice was low, but showing the blood data was only intended to verify that neutrophil depletion had worked systemically in these animals. Since this was validated, the data from individual tumors (n= 8-13 per group) from the same Figure panel can be analyzed. Of note, we applied the same protocol for neutrophil depletion, using a double antibody approach, which we had developed and validated in our previous study (Boivin *et al.*, Nat Com 2020, PMID 32488020). Thus, many more mice have received this antibody combination, which consistently reduces blood neutrophil counts by 80-90%.

To monitor better the effects of A-133 on neutrophil mobilization, we have now performed an A-133 daily injection for a short period of 3 days in healthy mice and have collected immune cells from blood. Agreeing with the trend observed in Fig. 4F in tumor-bearing mice, in healthy mice we detect an increased proportion of blood neutrophils in response to A-133. This agrees with data obtained by REF Levenson *et al.* when treating rats with the same molecule. Thus, this Bcl-xL blocker does not cause neutropenia; rather, it may increase neutrophil mobilization. The data on healthy mice are pasted below for the Reviewer:

3-days A-1331852 increases the proportion of blood neutrophils. Percentage of neutrophils in blood in control (n=5) and A-1331852-treated mice (n=6). Data are shown as mean \pm S.D. Significance was determined by the Mann-Whitney test.

Minor Points

1. Lines 298 and 303. Please insert "Ab" after "anti-rat".

“antibodies” have been added accordingly (now p.15).

2. Many references are incomplete. Volume and page numbers should be presented.

We are sorry for this and have now completed all references appropriately.

3. Fig. 2A and B lacks statistical comparisons.

We apologize for having missed this. Statistics with exact p-values have now been added directly on these two Figure panels: Fig. 2A and 2B.

4. Fig. 6E. Viability is expressed relative to medium only. Thus, a 3-fold increase is difficult to interpret in A549 SN only. Expressing viability in percentage should be more informative.

We have now expressed viability in percentage following the Reviewer's advice (new Fig. 6E).

Referee #3 (Comments on Novelty/Model System for Author):

The technical quality of the experiments displayed is adequate. The authors perform the necessary controls and the endpoints chosen are reasonable. I think the novelty of this manuscript is high. Neutrophils are known to be plastic in the context of cancer and it is appreciated that some tumor associated neutrophils promote tumor progression. However, details regarding which neutrophil subsets are tumor-promoting are still uncertain and the biology of these neutrophils is an active area of research. This study provides new insight into TANs and TAN biology. Further it provides a feasible/attractive strategy to selectively target tumor-promoting TANs. Thus, the potential impact is also high. The studies exploit a GEMM of NSCLC and human tissue, these are adequate models to address the major queries of the study.

Referee #3 (Remarks for Author):

Bodac et al present a study that nominates targeting Bcl-xlhigh tumor associated neutrophils (TANs) as an intriguing novel therapeutic strategy for lung cancer. The introduction section provides an effective framework for the study. TAN biology is an area of active investigation by many groups. Queries of interest include: the heterogeneity of neutrophil phenotypes in tumors, whether distinct subsets of TANs can be associated with tumor promotion and how to effectively and safely target tumor promoting TANs. Bodac and colleagues provide potential insight into these questions. The authors note that some neutrophils survive longer in tumors than anticipated and that there are conditions/pathologies where neutrophil apoptosis is delayed.

These observations induce the authors to investigate the expression of genes known to modulate survival or apoptosis. This results in identification of Bcl2l1 (Bcl-xl) as a gene/protein with elevated expression in TANs compared to neutrophils in healthy lung (HLN). The authors subsequently provide evidence that Bcl-xl expression can be induced by GM-CSF produced by tumor cells and that GM-CSF induction of Bcl-xl is dependent upon JAK/STAT activity. They also provide evidence that Bcl-xl expression correlates with neutrophil expression of SiglecF, a marker that has been associated with tumor-promoting neutrophils. The authors present the surprising finding that selective inhibition of Bcl-xl with A-1331852 reduces the viability of TANs and SiglecF+ neutrophils but not bone marrow-derived neutrophils. The authors find that treatment of mice bearing KP lung tumors with A-1331852 reduces tumor growth and reduces the number of aged neutrophils in the tumor microenvironment. Thus nominating targeting aged TANs by inhibiting Bcl-xl as a novel strategy for lung cancer.

The manuscript is presented clearly and the data displayed in general support the conclusions drawn. The comments below are largely minor in nature.

Comments:

1. HLN express higher levels of Bcl-xl than neutrophils in the bone marrow or blood. Do HLN survive longer than neutrophils in other organs/systems?

We thank the Reviewer for the comments and suggestions. To respond to this first comment, we have performed a BrdU injection in healthy mice to monitor the presence of young (2.5 days after BrdU) or old (5 days) neutrophils. We collected these cells from blood, kidney and lung. Our data show ~40% BrdU+ neutrophils in all tested sites at 2.5 days after injection, as expected. They also reveal a decreased proportion of BrdU+ neutrophils in blood, kidney and lung between the two time points, with the sharpest decrease in blood. We did not detect any significant difference in the abundance of BrdU+ neutrophils at 5 days post-BrdU injection when comparing lung and kidney, and thus conclude that HLNs do not survive longer compared to neutrophils homing to the kidney. A similar experiment (with 2.5 and 6.5 days BrdU time points) had been performed by us to compare TANs, HLNs, blood neutrophils, splenic neutrophils and bone marrow neutrophils. In that case, we had observed a slightly more elevated abundance of BrdU+ HLNs at 6.5 days compared to neutrophils from bone marrow, blood or spleen. This fits with what we observe between HLNs and blood neutrophils at 5 days. We conclude that neutrophil lifespan from lung and kidney does not differ significantly, but is longer compared to that in spleen, bone marrow and blood. Unfortunately, we failed to isolate neutrophils from two other organs, brain and intestine (see below the near absence of CD11b+Ly6G+ cells detected in these organs). The published data can be found in *Cancer Res.* 2021 May 1;81(9):2345-2357, *Figure 4C*, and the new data are pasted below for the Reviewer:

5 days-old neutrophils can be found in lung and in kidney. (A) 10 mice were injected with BrdU. For 5 mice, tissue was collected 2.5 days after injection and for the other 5, after 5 days. Flow cytometry gating strategy showing neutrophils (Cd11b⁺Ly6G⁺) in different tissues. Below are the representative histograms showing BrdU signal in neutrophils. (B) Quantification of the percentage of BrdU⁺ neutrophils. Data are shown as mean \pm S.D.

2. The flow data displayed in Figure 3&4 is convincing that treatment of A-1331852 alters the immune landscape. Yet the authors note that combining A-1331852 with anti-PD-1 was not an effective combination for controlling tumor progression (SFig5). Is there an observable change in T cells (Cd8 & Cd4) and Tregs in terms of location in the tumor after treatment with A-1331852, I wonder if multi-plex IHC on tumor sections might reveal that while T cell numbers were increased, effector T cells are still excluded from tumor cell nests, thus potentially explaining in part the lack of effective combo therapy?

We thank the Reviewer for raising this interesting point. As Reviewer is mentioning, most T cells reside adjacent to the KP tumors, often in tertiary lymphoid-like structures in this model, and a lack of T cell re-localization to the tumor bed might help explain the lack of effect of the combination therapy compared to A-133 alone. To address this point, we have performed a multiplex-IF of tumor sections from control and A-133-treated mice, co-staining with anti-CD8, anti-CD4, anti-FoxP3 (Treg), pan-cytokeratin (pan-CK, tumor epithelial cells) and DAPI. As the Reviewer had anticipated, upon A-133 treatment we fail to detect a re-localization of T cells into tumor nests, their localization still being outside the tumor: at its periphery and sometimes clustered in TLSs or similar immune cell aggregates found adjacent to tumors. We show in **new Appendix Figure S2D** a few representative examples of this staining. The text has been updated on p.16. Furthermore, we discuss the absence of lymphocyte tumor homing as possible explanation for the lack of sensitization of A-133 to anti-PD1 in the text, Discussion chapter, p.19.

14th Nov 2023

Dear Prof. Meylan,

Thank you for submitting your revised manuscript. We have now received the reports from the three referees who re-reviewed your manuscript, and as you will see below, they are supportive of publication pending minor revisions. I will therefore be able to accept your manuscript once the following points will be addressed:

1/ Referees' comments:

- Please address the remaining concerns from referee #2. The decision to include the additional figures in the main manuscript (or supplementary figures) is left to your discretion.

2/ Manuscript text:

- Please remove the blue highlights and only keep in track changes any new modification.

- Authors: regarding the addition of author Sarika Rana, please confirm that all authors were notified and agreed to it.

- Please provide up to 5 keywords.

- Please remove Data not shown (p.10): As per our guidelines on "Unpublished Data", the journal does not permit citation of "Data not shown". All data referred to in the paper should be displayed in the main or Expanded View figures.

- Materials and methods:

o Animals: please provide gender and age of the mice at time of experiment.

o Antibodies: please provide dilutions/concentrations for all antibodies.

o Human material: please add a statement that the experiments conformed to the principles set out in the WMA Declaration of Helsinki and the Department of Health and Human Services Belmont Report and correct the checklist accordingly.

o Statistics: please include a statement about blinding (even if no blinding was done) and about randomization and correct the checklist accordingly.

- The sentence about BioRender should be included in a "Graphics" subsection of the Material and Methods:

Graphics:

(some of the... OR Figure #... OR synopsis) Graphics were created with BioRender.com.

- Please add a "Disclosure statement and competing interests" statement: We updated our journal's competing interests policy in January 2022 and request authors to consider both actual and perceived competing interests. Please review the policy <https://www.embopress.org/competing-interests> and update your competing interests if necessary.

2/ Figures and Appendix:

- Figure legends:

1. Please indicate the statistical test used for data analysis in the legends of figures EV1a-b.

2. Please note that information related to n is missing in the legends of figures 6e; EV1b; EV2d; EV3b-c, h; EV5a-b.

3. Please indicate what white arrows represent in the legend of figure 6b

- Appendix: Please add page numbers in the appendix and table of content. Legends are missing for Fig. S1 and Table S1. The nomenclature needs correcting to "Appendix Table S1" and Appendix Figure S1" etc. throughout. We would advise you to format Table S1 to be portrait-oriented like Table S3.

4/ Source Data:

Thank you for providing Source Data. Regarding the deposition of microscopy images and cytometry files, I would encourage you to contact our Source Data coordinator, Hannah Sonntag, at h.sonntag@source-data.org.

5/ Synopsis:

Thank you for providing a nice synopsis image. Please upload it as a separate png/jpeg/tiff file 550 pixels wide x 250-500 pixels high.

6/ As part of the EMBO Publications transparent editorial process initiative (see our Editorial at

<http://embomolmed.embopress.org/content/2/9/329>), EMBO Molecular Medicine will publish online a Review Process File (RPF) to accompany accepted manuscripts.

This file will be published in conjunction with your paper and will include the anonymous referee reports, your point-by-point response and all pertinent correspondence relating to the manuscript. Let us know whether you agree with the publication of the RPF and as here, if you want to remove or not any figures from it prior to publication.

I look forward to receiving your revised manuscript.

Yours sincerely,

Lise Roth

***** Reviewer's comments *****

Referee #1 (Remarks for Author):

The authors have addressed all my comments

Referee #2 (Comments on Novelty/Model System for Author):

The new data and the revision lend additional support to the authors' suggestion of selective targeting of Siglec+, Bcl-xL-expressing tumor-associated neutrophils to control the growth of lung carcinoma in a mouse model. The manuscript sheds further light into neutrophil plasticity and raises the exciting possibility of therapeutic targeting of tumor-promoting neutrophil subset(s) without compromising host defense. I have few minor points that may require further clarification.

Referee #2 (Remarks for Author):

The authors have carefully addressed all of my previous concerns. The new data with intermittent treatment with A-1331852 and assessment of Bcl-2 family protein profile in particular have considerably strengthened the manuscript. There are few minor issues that may require further attention.

1. The findings with Intermittent treatment with A-1331852 (selective depletion of Siglec+ TANs without affecting total TANs) are important and lend strong support to the authors' concept. The authors may wish to consider moving these results into the main body of the manuscript.
2. The observation of A-1331852-evoked increases in the proportion of neutrophil (as shown in the rebuttal) is also relevant, and should be included as a supplemental figure.
3. Molecular weight markers are missing in all western blots.

Referee #3 (Comments on Novelty/Model System for Author):

No concerns

Referee #3 (Remarks for Author):

The authors have addressed the prior concerns effectively. I think this is an interesting study that has impact.

1/ Referees' comments:

- Please address the remaining concerns from referee #2. The decision to include the additional figures in the main manuscript (or supplementary figures) is left to your discretion.

2/ Manuscript text:

- Please remove the blue highlights and only keep in track changes any new modification.

The highlights have been removed, and the new modifications kept in track changes.

- Authors: regarding the addition of author Sarika Rana, please confirm that all authors were notified and agreed to it.

We confirm this.

- Please provide up to 5 keywords.

Keywords have been provided below the Abstract.

- Please remove Data not shown (p.10): As per our guidelines on "Unpublished Data", the journal does not permit citation of "Data not shown". All data referred to in the paper should be displayed in the main or Expanded View figures.

This has been removed.

- Materials and methods:

o Animals: please provide gender and age of the mice at time of experiment.

This has been added.

o Antibodies: please provide dilutions/concentrations for all antibodies.

This has been added.

o Human material: please add a statement that the experiments conformed to the principles set out in the WMA Declaration of Helsinki and the Department of Health and Human Services Belmont Report and correct the checklist accordingly.

This statement has been added in the Methods and corrected in the checklist.

o Statistics: please include a statement about blinding (even if no blinding was done) and about randomization and correct the checklist accordingly.

This sentence was moved from the "Mouse treatments" to the "Statistics" subsection.

- The sentence about BioRender should be included in a "Graphics" subsection of the Material and Methods:

Graphics:

(some of the... OR Figure #... OR synopsis) Graphics were created with BioRender.com.

A "Graphics" subsection has been added with the corresponding information.

- Please add a "Disclosure statement and competing interests" statement: We updated our journal's competing interests policy in January 2022 and request authors to consider both actual and perceived competing interests. Please review the policy <https://www.embopress.org/competing-interests> and update your competing interests if necessary.

This statement has been added after the Acknowledgements.

2/ Figures and Appendix:

- Figure legends:

1. Please indicate the statistical test used for data analysis in the legends of figures EV1a-b.

This has been added in the Legend.

2. Please note that information related to n is missing in the legends of figures 6e; EV1b; EV2d; EV3b-c, h; EV5a-b.

This has been added in the Legend.

3. Please indicate what white arrows represent in the legend of figure 6b

This has been added in the Legend.

Referee #1 (Remarks for Author):

The authors have addressed all my comments

Referee #2 (Comments on Novelty/Model System for Author):

The new data and the revision lend additional support to the authors' suggestion of selective targeting of Siglec+, Bcl-xL-expressing tumor-associated neutrophils to control the growth of lung carcinoma in a mouse model. The manuscript sheds further light into neutrophil plasticity and raises the exciting possibility of therapeutic targeting of tumor-promoting neutrophil subset(s) without compromising host defense. I have few minor points that may require further clarification.

Referee #2 (Remarks for Author):

The authors have carefully addressed all of my previous concerns. The new data with intermittent treatment with A-1331852 and assessment of Bcl-2 family protein profile in particular have considerably strengthened the manuscript. There are few minor issues that may require further attention.

1. The findings with Intermittent treatment with A-1331852 (selective depletion of Siglec+ TANs without affecting total TANs) are important and lend strong support to the authors' concept. The authors may wish to consider moving these results into the main body of the manuscript.

We thank Reviewer for this suggestion. However, Figure EV4 is entirely dedicated to these new results, highlighting them well. Considering that Main Figure 4 is already packed, we thus prefer to keep the new results with the intermittent treatment where they are.

2. The observation of A-1331852-evoked increases in the proportion of neutrophil (as shown in the rebuttal) is also relevant, and should be included as a supplemental figure.

According to Reviewer's suggestion, we have added this new data showing increased blood neutrophil proportions after 3-days A-1331853 treatment in healthy mice, as **new Figure 4G**, immediately following panel 4F that shows increased blood neutrophil proportions after 3-weeks A-1331853 treatment in tumor-bearing mice. The text has been adapted accordingly.

3. Molecular weight markers are missing in all western blots.

We thank Reviewer for noticing this and have now added the molecular weight markers for all Western blots (Fig. 1H, 2D and EV3C).

Referee #3 (Comments on Novelty/Model System for Author):

No concerns

Referee #3 (Remarks for Author):

The authors have addressed the prior concerns effectively. I think this is an interesting study that has impact.

28th Nov 2023

Dear Prof. Meylan,

I am pleased to inform you that your manuscript is accepted for publication and is now being sent to our publisher to be included in the next available issue of EMBO Molecular Medicine!

Your manuscript will be processed for publication by EMBO Press. It will be copy edited and you will receive page proofs prior to publication. Please add the link to the data deposited in FlowRepository at this stage.

Please note that you will be contacted by Springer Nature Author Services to complete licensing and payment information.

If you have any questions, please do not hesitate to contact the Editorial Office. Thank you for your contribution to EMBO Molecular Medicine, and congratulations on your interesting work!

Yours sincerely,

Lise Roth
